# PRIMAL-DUAL CONTINUAL LEARNING: STABILITY AND PLASTICITY THROUGH LAGRANGE MULTIPLIERS

## ABSTRACT

Continual learning is inherently a constrained learning problem. The goal is to learn a predictor under a *no-forgetting* requirement. Although several prior studies formulate it as such, they do not solve the constrained problem explicitly. In this work, we show that it is both possible and beneficial to undertake the constrained optimization problem directly. To do this, we leverage recent results in constrained learning through Lagrangian duality. We focus on memory-based methods, where a small subset of samples from previous tasks can be stored in a replay buffer. In this setting, we analyze two versions of the continual learning problem: a coarse approach with constraints at the task level and a fine approach with constraints at the sample level. We show that dual variables indicate the sensitivity of the optimal value with respect to constraint perturbations. We then leverage this result to partition the buffer in the coarse approach, allocating more resources to harder tasks, and to populate the buffer in the fine approach, including only impactful samples. We derive sub-optimality bounds, and empirically corroborate our theoretical results in various continual learning benchmarks. We also discuss the limitations of these methods with respect to the amount of memory available and the number of constraints involved in the optimization problem.

## 1 INTRODUCTION

In real-world settings, agents need to adapt to a dynamic stream of observations they receive from the environment. This has led to a plethora of research in *continual learning*, where the goal is to train agents to solve a set of diverse tasks presented sequentially (Thrun & Mitchell, 1995).

Since the capacity of machine learning models is limited, the challenge in continual learning is balancing the acquisition of new knowledge (plasticity) and the consolidation of previously integrated knowledge (stability). A potential consequence of poorly handling the so-called stability-plasticity dilemma is severe performance degradation in past tasks. Avoiding this phenomenon—referred to as *catastrophic forgetting* (McCloskey & Cohen, 1989; French, 1999)—naturally leads to constrained optimization formulations, which have appeared extensively in the continual learning literature (Aljundi et al., 2019; Chaudhry et al., 2018; Lopez-Paz & Ranzato, 2017; Peng et al., 2023).

Most approaches do not solve this constrained optimization problem explicitly. Instead, they use gradient projections (Lopez-Paz & Ranzato, 2017; Chaudhry et al., 2018) or promote proximity in the parameter space (Wang et al., 2021b; Kirkpatrick et al., 2017). This work shows that it is both possible and beneficial to undertake the constrained learning problem directly (*Contribution 1*). To do this, we leverage recent advances in constrained learning through Lagrangian duality (Chamon & Ribeiro, 2020) and build a framework that contemplates both task-level and sample-level forgetting.

State-of-the-art continual learning methods tend to include replay buffers, in which agents store a small subset of the previously seen instances. These methods have become ubiquitous, since they generally outperform their memoryless counterparts (Masana et al., 2022; Zhou et al., 2023; De Lange et al., 2021). The *principled* constrained learning framework proposed in this paper enables an adaptive and efficient management of the memory buffer.

Specifically, we first show that Lagrangian dual variables resulting from the proposed primal-dual method capture the stability-plasticity trade-off, since they indicate the sensitivity of the optimal value with respect to constraint perturbations (*Contribution 2*). At the task level, we leverage this

result to partition the buffer, allocating more resources to harder tasks; and at the sample level, we use it to populate the buffer, including only impactful samples (*Contribution 3*). These techniques give us a direct handle on the stability-plasticity trade-off incurred in the learning process.

We showcase the benefits of the proposed method for several memory budgets in a diverse set of continual learning benchmarks, including image, audio, and medical datasets. We also study the sensitivity to the forgetting tolerance allowed and discuss the limitations of the proposed primal-dual method with respect to the number of constraints, the sampling of outliers, and the underestimation of task difficulties.

## 2 CONTINUAL LEARNING IS A CONSTRAINED LEARNING PROBLEM

In continual learning, the goal is to learn a predictor that minimizes the expected risk over a set of *tasks*,

$$f^\star = \arg\min_{f \in \mathcal{F}} \sum_{t=1}^{T} \mathbb{E}_{\mathfrak{D}_t} \left[ \ell(f(x), y) \right],$$

where $T$ is the number of tasks, $\mathfrak{D}_t$ is the data distribution associated to task $t$ and $\mathcal{F}$ is a functional space. The tasks and their corresponding data distributions are observed sequentially. That is, at time $t$, data from previous tasks (i.e., $\mathfrak{D}_1, \cdots, \mathfrak{D}_{t-1}$) and from future tasks (i.e., $\mathfrak{D}_{t+1}, \cdots, \mathfrak{D}_T$) are not available. In this setting, the main issue that arises is catastrophic forgetting: if we sequentially fine-tune $f$ on each incoming distribution, the performance on previous tasks could drop severely. A continual learner is one that is stable enough to retain acquired knowledge and malleable enough to gain new knowledge.

If the no-forgetting requirement is enforced at the task level, we can formulate the continual learning problem as minimizing the statistical risk on the current task without harming the performance of the model on previous tasks, i.e.,

$$P_t^\star = \arg\min_{f \in \mathcal{F}} \ \mathbb{E}_{\mathfrak{D}_t}[\ell(f(x), y)], \qquad\qquad (P_t)$$

$$\text{s.t.} \quad \mathbb{E}_{\mathfrak{D}_k}[\ell(f(x), y)] \le \epsilon_k, \quad \forall\, k \in \{1, \ldots, t-1\},$$

where $\epsilon_k \in \mathbb{R}$ is the *forgetting tolerance* of task $k$, i.e., the worse average loss that is admissible in a certain task. In many cases, this is a *design* requirement, and not a tunable parameter. For instance, in medical applications, $\epsilon_k$ can be tied to regulatory constraints. If the upper bound is set to the unconstrained minimum (i.e., $\epsilon_k = \min_{f \in \mathcal{F}} \mathbb{E}_{\mathfrak{D}_k}[\ell(f(x), y)]$), then we are implementing an *ideal continual learner* (Peng et al., 2023). However, we do not have access to $\mathfrak{D}_k$ for $k \ne t$, but only to a *memory buffer* $\mathcal{B}_t = \cup_{k=1}^{t-1} \mathcal{B}_t^k$, where $\mathcal{B}_t^k$ denotes the subset of the buffer allocated to task $k$ while observing task $t$. When possible, we will obviate the dependence on the index $t$ to ease the notation.

In this setting, the main questions that arise are: (i) When is the constrained learning problem ($P_t$) solvable? (ii) How to solve it? (iii) How to partition the buffer $\mathcal{B}$ across the different tasks ? (iv) Which samples from each task should be stored in the buffer?

This paper is structured as follows: in Section 3, we present the Lagrangian duality framework used to undertake the constrained learning problem. In Section 4, we turn our attention to the buffer partition strategy, and in Section 5 we discuss the dual variable-based approach to sample selection.

### 2.1 SETTING

For continual learning to be justified, tasks need to be similar. The following assumptions characterize this similarity in terms of the distance between sets of optimal predictors.

**Assumption 2.1** (*Task Similarity*): *Let $\mathcal{F}_t^\star = \{f \in \mathcal{F} : \mathbb{E}_{\mathfrak{D}_t}[\ell(f(x), y)] = \min_f \mathbb{E}_{\mathfrak{D}_t}[\ell(f(x), y)]\}$ be the set of optimal predictors associated to task $t$. The pairwise distance between these sets across different tasks is upper-bounded by a constant $\delta > 0$, i.e.,*

$$d(\mathcal{F}_i^\star, \mathcal{F}_j^\star) \le \delta, \quad \forall i, j \in \{1, \cdots, T\}.$$

Several task similarity assumptions have been proposed in the literature, most of which can be formulated as Assumption 2.1 with an appropriate choice of $d(\cdot, \cdot)$ and $\delta$. In

this work, we use the standard (Haussdorf) distance between non-empty sets: $d(X, Y) = \max \left\{ \sup_{x \in X} d(x, Y), \sup_{y \in Y} d(X, y) \right\}$. In over-parameterized settings, deep neural networks attain near-interpolation regimes and this assumption is not strict (Liu et al., 2021).

**Assumption 2.2** *(Constraint Qualification): The loss $\ell$ and functional space $\mathcal{F}$ are convex, and there exists a strictly feasible solution (i.e., $\exists\, f \in \mathcal{F}$ such that $\mathbb{E}_{\mathfrak{D}_k}[\ell(f(x), y)] < \epsilon_k, \forall k$).*

Note that convexity is assumed with respect to function $f$, not model parameters, and is satisfied by typical losses, such as mean-squared error and cross-entropy loss. We will consider that the functional space $\mathcal{F}$ is endowed with the $L_2$ norm, and we also consider a parameterization (e.g., a neural network) $\boldsymbol{\Theta}$, such that $\mathcal{F}_{\boldsymbol{\Theta}} = \{f_{\boldsymbol{\theta}} : \boldsymbol{\theta} \in \boldsymbol{\Theta}\} \subseteq \mathcal{F}$.

**Assumption 2.3** *(Near-Universality of the Parameterization): For all $f \in \mathcal{F}$, there exists $\boldsymbol{\theta} \in \boldsymbol{\Theta}$ such that for a constant $\nu > 0$, we have $\|f - f_{\boldsymbol{\theta}}\|_{L_2} \leq \nu$.*

The near-universality assumption is directly related to the richness of the parameterization (or model capacity). In over-parameterized models, such as deep neural networks, this assumption is expected to hold with a small $\nu$.

**Assumption 2.4** *(Uniform Convergence): There exists $R > 0$ such that $\|f\|_{L_2} \leq R$ for every $f \in \mathcal{F}$ and the loss $\ell(f(x), y)$ is $M$-Lipschitz in $f$.*

This assumption is standard in statistical learning (Shalev-Shwartz et al., 2009) and guarantees uniform convergence.

## 3 CONTINUAL LEARNING IN THE DUAL DOMAIN

The following proposition sheds light on the dependence between the forgetting tolerance $\epsilon_k$ and the task similarity magnitude $\delta$.

**Proposition 3.1** *Let $m_k$ be the unconstrained minimum associated to task $k$. Under Assumptions 2.1 and 2.4, $\exists\, f \in \mathcal{F}$ such that,*

$$\mathbb{E}_{\mathfrak{D}_k}[\ell(f(x), y)] \leq m_k + \frac{T-1}{T} M\delta, \quad \forall k \in \{1, \cdots, T\}. \tag{1}$$

Proposition 1 suggests that for Problem ($P_t$) to be solvable, the forgetting tolerances $\{\epsilon_k\}$ need to match the task similarity $\delta$. For instance, if $\epsilon_k = m_k + M\delta$ for all $k$, then problem ($P_t$) is feasible at all iterations, and its solution is $M\delta$ close to the optimum in terms of the expected loss on the current task. In what follows, we explain how to undertake this constrained learning problem once the forgetting tolerances are set.

As done in standard supervised learning, to solve problem ($P_t$), the function class $\mathcal{F}$ is parameterized (e.g., by a neural network), and expectations are approximated by sample means. ($P_t$) is a statistical constrained optimization problem, whose Lagrangian empirical dual can be written as

$$\hat{D}_t^\star = \max_{\boldsymbol{\lambda} \in \mathbb{R}_+^t} \min_{\boldsymbol{\theta} \in \boldsymbol{\Theta}} \hat{\mathcal{L}}(\boldsymbol{\theta}, \boldsymbol{\lambda}) := \frac{1}{n_t} \sum_{i=1}^{n_t} [\ell(f_{\boldsymbol{\theta}}(x_i), y_i)] + \sum_{k=1}^{t} \lambda_k \left( \frac{1}{n_k} \sum_{i=1}^{n_k} [\ell(f_{\boldsymbol{\theta}}(x_i), y_i)] - \epsilon_k \right), \tag{$\hat{D}_t$}$$

where $\hat{\mathcal{L}}(\boldsymbol{\theta}, \boldsymbol{\lambda})$ denotes the empirical Lagrangian, $n_k$ denotes the number of samples from task $k$ available at iteration $t$, and $\boldsymbol{\lambda} = [\lambda_1 \; \ldots \; \lambda_t]^T$ denotes the set of dual variables corresponding to the task-level constraints. For a fixed $\boldsymbol{\lambda}$, the Lagrangian $\hat{\mathcal{L}}(\boldsymbol{\theta}, \boldsymbol{\lambda})$ is a regularized objective, where the losses on previous tasks act as regularizing functionals. Thus, the saddle point problem in ($\hat{D}_t$) can be viewed as a two-player gamer, or as a regularized minimization, where the regularization weight $\boldsymbol{\lambda}$ is updated during the training procedure according to the degree of constraint satisfaction or violation. This contrast with several replay and knowledge distillation approaches that augment the loss using manually-tuned hyperparameters (Buzzega et al., 2020a; Michieli & Zanuttigh, 2021).

In general, the dual problem yields a lower bound on $P_t^\star$, which is known as *weak duality*. However, under certain conditions, $D_t^\star$ attains $P_t^\star$ and the optimal dual variable $\boldsymbol{\lambda}^\star$ indicates the sensitivity

of $P_t^\star$ with respect to constraint perturbations (Rockafellar, 1997). More precisely, we state the following theorem, whose proof can be found in Appendix A.2, which characterizes the variations of $P_t^\star$ as a function of the constraint levels $\{\epsilon_k\}_{k=1}^t$, and serves as a motivation for the proposed memory partition and sample selection methods.

> **Theorem 3.2** *Under Assumption 2.2, we have*
>
> $$-\lambda_k^\star \in \partial P_t^\star(\epsilon_k), \; \forall k \in \{1, \ldots, t\}, \tag{2}$$
>
> *where $\partial P_t^\star(\epsilon_k)$ denotes the sub-differential of $P_t^\star$ with respect to $\epsilon_k$, and $\lambda_k^\star$ is the optimal dual variable associated to the constraint on task $k$.*

Provided we have enough samples per task and the parameterization is rich enough, $(\hat{D}_t)$ can approximate the constrained statistical problem $(P_t)$. More precisely, the empirical duality gap, defined as the difference between the optimal value of the empirical dual and the statistical primal, is bounded (Chamon & Ribeiro, 2020). Furthermore, the dual function $g_p(\lambda) = \min_{\theta \in \Theta} \hat{\mathcal{L}}(\theta, \lambda)$ is the minimum of a family of affine functions on $\lambda$, and thus is concave. Consequently, the outer problem corresponds to the maximization of a concave function and can be solved via sub-gradient ascent (Nedić & Ozdaglar, 2009). The inner minimization, however, is generally non-convex, but there is ample empirical evidence that deep neural networks can attain *good* local minima when trained with stochastic gradient descent (Zhang et al., 2016). The max-min problem $(\hat{D}_t)$ can be undertaken by alternating the minimization with respect to $\theta$ and the maximization with respect to $\lambda$ (K. J. Arrow & Uzawa, 1960). We elaborate on this when we present the algorithm in Section 4.

## 4 OPTIMAL BUFFER PARTITION

### 4.1 DUAL VARIABLES CAPTURE THE STABILITY-PLASTICITY TRADE-OFF

In Section 3, we argued that continual learning can be tackled in the dual domain, resulting in primal and dual variables $f^\star$ and $\lambda^\star$. Throughout the analysis, we treated the number of samples per task in the buffer $\{n_k\}_{k=1}^t$ as fixed parameters. However, we can also treat them as optimization variables, leading to a memory partition strategy. Different tasks have different intrinsic difficulties and sample complexities. Thus, random or uniform partitions are typically sub-optimal.

Theorem 3.2 implies that for any task $k$, $-\lambda_k^\star$ yields a global linear under-estimator of $P_t^\star$ at $\epsilon_k$, i.e., for any $\gamma \in \mathbb{R}$,

$$P_t^\star(\epsilon_k + \gamma) - P_t^\star(\epsilon_k) \geq \langle -\lambda_k^\star, \gamma \rangle. \tag{3}$$

This means that the optimal dual variable $\lambda_k^\star$ carries information about the difficulty of task $k$. Specifically, tightening the constraint associated to task $k$ ($\gamma < 0$) would restrict the feasible set, causing a degradation of the optimal value of $(P_t)$ at a rate larger than $\lambda_k^\star$. That is, optimal dual variables reflect how hard it is to achieve good performance in the current task (*plasticity*), while maintaining the performance on a previous task (*stability*). Therefore, $\lambda_k^\star$ captures the stability-plasticity trade-off associated to task $k$.

In light of this result, it is sensible to partition the buffer across different tasks as an increasing function of $\lambda^\star$, allocating more resources to tasks with higher associated dual variables. In what follows, we propose an approach that leverages the information provided by $\lambda^\star$ and also contemplates the Lagrangian generalization gap.

### 4.2 MEMORY PARTITION AND GENERALIZATION

Assumption 2.4 implies that for any $\delta \in (0, 1)$ and $f \in \mathcal{F}$, with probability at least $1 - \delta$, we have

$$\left| \mathbb{E}_{\mathfrak{D}_k} [\ell(f(x), y)] - \frac{1}{n_k} \sum_{i=1}^{n_k} \ell(f(x_i), y_i) \right| \leq \zeta(n_k), \; \forall k \in \{1, \ldots, t\}, \tag{4}$$

where $\zeta(n_k, \delta) = O\left( \frac{RM\sqrt{d\log(n_k)\log(d/\delta)}}{\sqrt{n_k}} \right)$ approaches zero as the sample size $n_k$ goes to infinity (Shalev-Shwartz et al., 2009, Theorem 5). Applying this bound, the generalization gap associated

---

**Algorithm 1** Primal-Dual Continual Learning (PDCL)

---

1: **Input:** Number of tasks $T$, datasets $\{\mathfrak{D}_t\}_{t=1}^T$, primal learning rate $\eta_p$, dual learning rate $\eta_d$, constraint levels $\{\epsilon_t\}_{t=1}^T$, number of iterations $n_{\text{iter}}$.
2: Initialize: $\boldsymbol{\theta}$
3: **for** $t = 1, \ldots, T$ **do**
4:      Initialize: $\boldsymbol{\lambda} \leftarrow \mathbf{0}$
5:      **for** $i = 1, \ldots, n_{\text{iter}}$ **do**
6:          $\boldsymbol{\theta} \leftarrow \boldsymbol{\theta} - \eta_p \nabla_{\boldsymbol{\theta}} \mathcal{L}(\boldsymbol{\theta}, \boldsymbol{\lambda})$  $(\times T_p)$   `// Update primal variables` ($T_p$ `SGD steps`)
7:          $s_k \leftarrow \frac{1}{n_k} \sum_{j=1}^{n_k} \ell(f_\theta(x_j), y_j) - \epsilon_k, \quad k = 1, \ldots, t.$ `// Evaluate constraint slacks`
8:          $\lambda_k \leftarrow [\lambda_k + \eta_d s_k]_+, \quad k = 1, \ldots, t.$                   `// Update dual variables`
9:      **end for**
10:     $n_1^\star, \ldots, n_t^\star \leftarrow BP(\lambda_1^\star, \ldots, \lambda_t^\star)$             `// Compute optimal buffer partition`
11:     $\mathcal{B}_t \leftarrow FB(\mathcal{B}_{t-1}, \mathfrak{D}_t, \{n_k^\star\}_{k=1}^t)$                          `// Fill Buffer`
12: **end for**
13: **Return:** $\boldsymbol{\theta}, \boldsymbol{\lambda}$.

---

with the Lagrangian can be written as:

$$\left| \sum_{k=1}^t \lambda_k \mathbb{E}_{\mathfrak{D}_k}[\ell(f(x), y)] - \sum_{k=1}^t \frac{\lambda_k}{n_k} \sum_{i=1}^{n_k} \ell(f(x_i), y_i) \right| \leq \sum_{k=1}^t \lambda_k \zeta(n_k). \tag{5}$$

where for task $t$, we replace $\lambda_t \leftarrow \lambda_t + 1$, as its loss appears in both the objective and the constraints. Therefore, we propose to find the buffer partition that minimizes this generalization gap by solving the following non-linear constrained optimization problem,

$$n_1^\star, \ldots, n_t^\star = \underset{n_1, \ldots, n_t \geq n_{\min}}{\arg\min} \quad \sum_{k=1}^t \lambda_k \, \zeta(n_k), \tag{BP}$$

$$\text{s.t.} \qquad \sum_{k=1}^t n_k = |\mathcal{B}|.$$

As explained in Section 4.1, the difficulty of a task can be assessed through its corresponding dual variable, since it captures the stability-plasticity tradeoff. In (BP), we minimize the sum of task sample complexities, weighting each one by its corresponding dual variable and restricting the total number of samples to the memory budget. We elaborate on how to solve this optimization problem and the role of $n_{\min}$ in Appendix A.8. An overview of the proposed primal-dual continual learning method (PDCL) is provided in Algorithm 1, where $FB$ represents a generic mechanism for populating the buffer with samples from the previously-observed tasks given a specific memory partition $\{n_1, \cdots, n_t\}$. As shown in Figure 1, when isolating the effect of the buffer partition, allocating more resources to tasks with higher dual variables is beneficial in terms of final mean error. In this experiment, we isolate the effect of buffer partition by comparing to Experience Replay (Rolnick et al., 2018) with Ring and Reservoir sampling (see Appendix A.1).

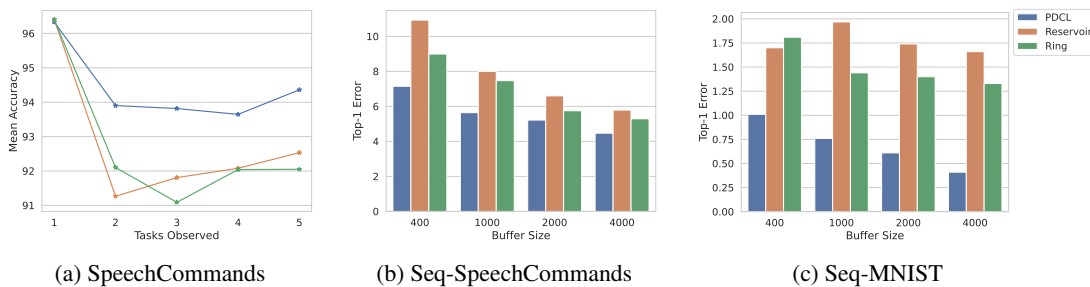

(a) SpeechCommands         (b) Seq-SpeechCommands         (c) Seq-MNIST

Figure 1: TIL performance of PDCL vs. two baseline memory partition methods on two image and audio datasets. Ring leads to a uniform partition and Reservoir approximates $\mathfrak{B}(x, y)$ to $\mathfrak{D}(x, y)$.

### 4.3 EMPIRICAL OPTIMAL DUAL VARIABLES

The sensitivity result in Theorem 3.2 holds for the optimal *statistical* dual variables $\lambda_u^\star$ of problem $(P_t)$. However, in practice, we have access to the *empirical* parameterized dual variables $\hat{\lambda}_p^\star$ of problem $(\hat{D}_t)$. In this section, we characterize the distance between these two quantities, showing that, under mild assumptions, $\hat{\lambda}_p^\star$ is not far from $\lambda_u^\star$. Let $g_p(\lambda) := \min_{\theta \in \Theta} \mathcal{L}(\theta, \lambda)$ and $g_u(\lambda) := \min_{f \in \mathcal{F}} \mathcal{L}(f, \lambda)$ denote the parameterized and unparameterized dual functions.

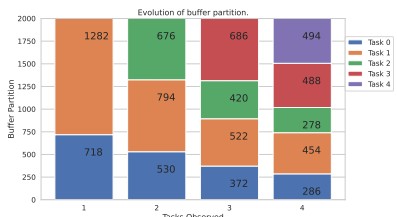

Figure 2: Evolution of buffer partition in SpeechCommands with $|\mathcal{B}| = 2000$.

**Proposition 4.1** *Under Assumptions 2.3 and 2.2, the point-wise distance between the parameterized and unparameterized dual functions is bounded by an affine function on $\|\lambda\|_1$,*

$$g_p(\lambda) - g_u(\lambda) \leq M\nu(1 + \|\lambda\|_1), \qquad \forall \lambda \succeq 0.$$

Optimal dual variables indicate the sensitivity of the optimal value with respect to constraint perturbations (see Section 3.2). Thus, the term $(1 + \|\lambda\|_1)$ can be seen as an indicator of the sensitivity of the optimization problem. Let $\mathcal{B}_\lambda$ denote the segment connecting $\lambda_u^\star$ and $\hat{\lambda}_p^\star$. The following theorem, whose proof can be found in Appendix A.3, captures the impact on optimal dual variables of approximating the expected values over $\mathfrak{D}_k$ by sample means.

---

**Theorem 4.2** *Let $c$ denote the strong concavity constant of $g_u(\lambda)$ in $\mathcal{B}_\lambda$. Under Assumptions 2.2, 2.3, and 2.4, with probability at least $1 - t\delta$, we have:*

$$\|\hat{\lambda}_p^\star - \lambda_u^\star\|_2^2 \leq \frac{2}{c} \left[ M\nu(1 + \|\hat{\lambda}_p^\star\|_1) + 6\zeta(\tilde{n}, \delta)(1 + \|\lambda'\|_1) \right],$$

*where $\|\lambda'\|_1 = \max\{\|\lambda_p^\star\|, \|\hat{\lambda}_p^\star\|\}$ and $\tilde{n} = \min_{i=1,\ldots,t} n_i$.*

---

The first term in this bound reflects the sub-optimality of $\hat{\lambda}_p^\star$ with respect to $\lambda_u^\star$, while the second term captures the effect of estimating expectations with sample means. We analyze the concavity constant $c$ in detail in Appendix A.5.

Theorem 4.2 implies that as the number of samples grows, and the capacity of the model increases (i.e., $\nu$ decreases), $\hat{\lambda}_p^\star$ approaches $\lambda_u^\star$. Thus, provided our model has enough capacity and the number of samples per task is large enough, $\hat{\lambda}_p^\star$ can be used as a sensitivity indicator of $P_t^\star$. A weak aspect of the bound in Theorem 4.2 is that the sample complexity that dominates it is the one associated with the task with the least number of samples. This can be fixed by replacing the minimum with the average sample complexity, but we pay the price of having the bound grow linearly with the number of tasks.

## 5 IMPACTFUL SAMPLE SELECTION

When filling the buffer with random samples from each distribution, there is no sampling bias (i.e., $\mathfrak{B}_k(x, y) = \mathfrak{D}_k(x, y)$), and the solution of $(P_t)$ has the no-forgetting guarantees from statistical constrained learning (Peng et al., 2023). However, performing sample selection can be beneficial due to the following reasons:

- The *i.i.d.* assumption may not hold, in which case sample selection has theoretical and empirical benefits, particularly as an outlier detection mechanism (Sun et al., 2022; Peng et al., 2023; Borsos et al., 2020).

- Random sampling is not optimal in terms of expected risk decrease rate, which is the main property exploited in active and curriculum learning (Bengio et al., 2009; Gentile et al., 2022; Elenter et al., 2022).

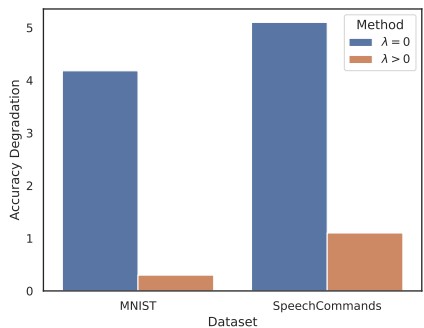
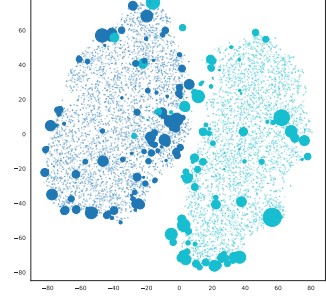

(a) Accuracy degradation on Task 1 when observing Task 2 and storing all samples with $\lambda > 0$ vs $\lambda = 0$.

(b) t-SNE map of classes 0 and 1 of MNIST, with associated dual variable indicated by the size of the marker, after iteration 2.

Figure 3: Informativeness of dual variables for sample selection. In (b), samples with large associated dual variables tend to accumulate in the task decision boundary and edges of the class cluster, while (a) shows that storing these samples, as opposed to storing those in the center of the cluster, is beneficial in terms of forgetting.

## 5.1 IDENTIFYING IMPACTFUL SAMPLES

Instead of task-level constraints, one could enforce a no-forgetting requirement at the sample level. For a fixed tightness $\epsilon$, this constraint is stricter than the task-level constraint and will enable sample selection. The no-forgetting requirement can be written as:

$$\ell(f(x), y) \leq \epsilon(x, y), \quad \mathfrak{B}_t^k\text{-a.e.} \quad \forall\, k = 1, \cdots, t, \tag{6}$$

where $\mathfrak{B}_t^k$ is the distribution induced by sampling $\mathfrak{D}_k$ to fill the memory buffer at iteration $t$. As explained in the beginning of Section 5, sampling non-randomly induces a bias in the buffer distribution: $\mathfrak{B}_t(x, y) \neq D_t(x, y)$. In what follows, we explore a dual variable-based sampling strategy that leverages the sensitivity of the optimization problem.

In this case, the update rule for the dual variables is given by

$$\lambda^{i+1}(x, y) = \left[\lambda^i(x, y) + \eta_d(\ell(f_{\lambda^i}(x), y) - \epsilon(x, y))\right]_+,$$

where $f_\lambda$ is the Lagrangian minimizer associated to $\lambda$, i.e: $f_\lambda = \arg\min_{f \in \mathcal{F}} \mathcal{L}(f, \lambda)$. Thus, in this formulation, dual variables accumulate the individual slacks over the entire learning procedure. This allows dual variables to be used as a measure of informativeness, while at the same time affecting the local optimum to which the algorithm converges. Similar ideas on monitoring the evolution of the loss—or training dynamics—for specific training samples in order to recognize impactful instances have been used in generalization analyses (Toneva et al., 2019; Katharopoulos & Fleuret, 2018) and active learning methods (Wang et al., 2021a; Elenter et al., 2022). In this case, a similar sensitivity analysis as in Section 3 holds at the sample level:

**Proposition 5.1** *Under Assumption 2.2, for all $(x, y)$:*

$$-\lambda_t^\star(x, y) \in \partial P_t^\star(\epsilon(x, y)), \tag{7}$$

*where $\partial P_t^\star(\epsilon(x, y))$ denotes the Fréchet subdifferential of $P_t^\star$ at $\epsilon(x, y)$.*

Proposition 5.1 implies that the constraint whose perturbation has the most potential impact on $P_t^\star$ is the constraint with the highest associated optimal dual variable. As in the task-level constraints, infinitesimally tightening the constraint in a neighborhood $(x, y)$ would restrict the feasible set, causing an increase of the optimal value of $P_t^\star$ at a rate larger than $\lambda(x, y)$. In that sense, the magnitude of the dual variables can be used as a measure of informativeness of a training sample. Similarly to non-support vectors in SVMs, samples associated to inactive constraints (i.e., $\{(x, y) : \lambda_t^\star(x, y) = 0\}$), are considered uninformative. This notion of informativeness is illustrated in Figure 3. As shown in

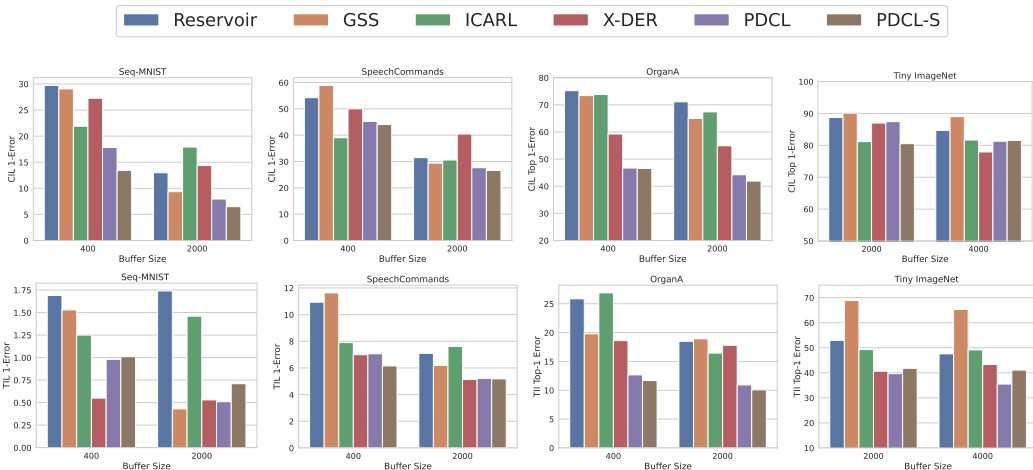

Figure 4: Error in the Class Incremental Learning (**top row**) and Task Incremental Learning (**bottom row**) settings for two different buffer sizes across four benchmarks (lower is better). Results for additional buffer sizes are presented in Appendix A.7.

the figure, large dual variables correspond to both outliers and inliers. Indeed, informative samples and outliers (such as mislabeled samples) may be hard to distinguish. Recent empirical findings indicate that many active and continual learning algorithms consistently prefer to acquire samples that traditional models fail to learn (Karamcheti et al., 2021).

In Primal-Dual Continual Learning with Sample selection (PDCL-S), the buffer is filled by leveraging the per-sample dual variables $\lambda(x, y)$. Specifically, given a buffer partition $n_1, \cdots, n_t$, the generic mechanism $FB$ for populating the buffer in Algorithm 1 is particularized to filling the buffer with the samples with the highest associated dual variable from each class. Thus, this method can be interpreted as a near-matching between the buffer-induced distribution $\mathcal{B}_t(x, y)$ and the optimal dual variable function $\lambda_t^\star(x, y)$. In order to avoid sampling outliers, we discard samples with extremely high dual variables before sampling.

## 6 EXPERIMENTAL VALIDATION

To highlight the versatility of the proposed approach, we evaluate it in four continual learning benchmarks, two image classification tasks (MNIST LeCun & Cortes (2010) and Tiny-ImageNet Le & Yang (2015)), one speech classification task (SpeechCommands Warden (2018)) and one medical (Abdominal CT Scan) dataset (OrganA Yang et al. (2021)). Each dataset is split into disjoint sets each containg a subset of the classes. MNIST, SpeechCommands, and OrganA are split into 5 tasks with 2 classes each. The more challenging task, Tiny ImageNet, is split into 10 tasks, each with 20 classes.

We adopt standard neural network architectures and match the model complexity to the difficulty of the problem at hand. In MNIST, we use a three-layer MLP with ReLU activations. In Seq. Speech-Commands and OrganA, we use four- and five-layer CNNs, respectively, with ReLU activations, Batch Normalization and MaxPooling. In TinyImagenet, we use a ResNet-18 architecture (He et al., 2016).

At each iteration $t$, models are trained using $f_{t-1}$ as initialization with a primal learning rate of $\eta_p = 0.001$ in all datasets except TinyImagenet, where $\eta_p = 0.01$ is used. The dual learning rate is set to $\eta_d = 0.05$ or $\eta_d = 0.5$ respectively. We adopt the baseline implementations of Mammoth[1], and use their reported hyperparameters for the baselines. We measure final average accuracy both in the Class Incremental Learning (CIL) and Task Incremental Learning (TIL) settings across 5 random seeds. More details about the forgetting tolerance parameter $\epsilon_k$ are presented in Section 6.1.

---

[1]https://github.com/aimagelab/mammoth

We evaluate both the memory partition (PDCL) and memory partition with sample selection (PDCL-S) methods, and compare their performance with the baseline approaches presented in Appendix A.1 that are most related to our work, namely Experience Replay (Rolnick et al., 2018) with Reservoir sampling, X-DER (Boschini et al., 2022), GSS (Aljundi et al., 2019) and iCARL (Rebuffi et al., 2016). Additional experimental details and results can be found in Appendix A.7.

Figure 6 compares the performance comparison of these continual learning methods in the CIL and TIL settings. We can observe that ndertaking the continual learning problem with a primal-dual algorithm, and leveraging the information provided by dual variables leads to comparatively low forgetting in almost all buffer sizes and benchmarks. It is important to note that sample selection does not always improve the performance of the method. This is consistent with previous works on the effectiveness of sample selection (Araujo et al., 2022), and with the fact that the datasets used do not have many outliers. Moreover, in settings such as CIL Tiny Imagenet, no method outperforms Reservoir by a significant margin, which is consistent with recent surveys (Zhou et al., 2023).

## 6.1 ABLATION ON THE FORGETTING TOLERANCE

As explained in Section 2, the forgetting tolerances $\{\epsilon_k\}_{k=1}^T$ correspond to the worst average loss that one requires of a past task. In many cases, this is a *design* requirement, and not a tunable parameter. For extremely large values of epsilon, the constraint slacks are always negative and dual variables quickly go to zero (analogous to an unconstrained problem), which makes them uniformative. On the other hand, extremely low values of $\epsilon_k$ might also be inadequate, since the tightness of these constraints can make the problem infeasible and make dual variables diverge. As shown in Figure 5, the method is not extremely sensitive to $\epsilon_k$ in the range $[0.15, 0.45]$. Our ablations suggest that values in the range $[1.05 m_k, 1.25 m_k]$, where $m_k$ is the average loss observed when training the model without constraints, work well in practice.

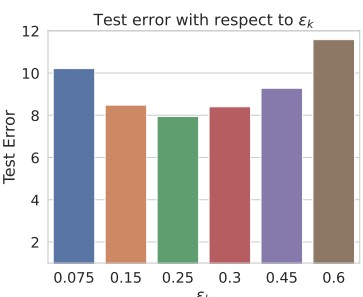

Figure 5: Ablation on the forgetting tolerance $\epsilon_k$ in Seq-MNIST.

## 7 DISCUSSION

In this work we presented a principled primal dual approach to continual learning that explicitly tackles learning under a no-forgetting requirement. We showed that dual variables play a key role in this framework, since they give us a handle on the stability-plasticity tradeoff by assessing the relative difficulty of a task and the impactfulness of a given sample.

One of the drawbacks exhibited by the proposed method is dual variable underestimation. It is possible that the difficulty of a task $k$ at a certain iteration $t_0$ is underestimated, and that the corresponding dual variable $\lambda_k$ re-grows at a future iteration $t_1$. This is an issue since we have already discarded the non-selected samples from task $k$, meaning that a portion of the buffer—characterized by $\lambda_k(t_1) - \lambda_k(t_0)$—would remain empty. To deal with this issue, one can fill the empty portion of the buffer with either: augmented samples from the previously selected ones or samples from the current task, whose dataset is entirely available. Another downside of our approach is that the number of constraints involved in the optimization problem can be very large, particularly when doing sample selection. This can increase the sensitivity of the optimization process to the learning rates and forgetting tolerances.

In this work, we have uniformly set the forgetting tolerances for all tasks or samples. However, a pretraining method that yields non-uniform, feasible, and informative constraint upper bounds could improve the performance of the proposed approach. Moreover, understanding the conditions under which sample selection is provably beneficial is also a promising direction for future work.

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

## A   APPENDIX

### A.1   RELATED WORK

Machine learning systems have become increasingly integrated into our daily lives, reaching critical applications from medical diagnostics (Kononenko, 2001) to autonomous driving (Kiran et al., 2021). Consequently, the development of machine learning models that can adapt to dynamic environments and data distributions has become a pressing concern. A myriad of strategies for continual learning, also referred to as lifelong or incremental learning, have been proposed in recent years (Ke et al., 2023; Guo et al., 2022; Aljundi et al., 2019; Chaudhry et al., 2018; Ermis et al., 2022; Rebuffi et al., 2016; Rolnick et al., 2018; Buzzega et al., 2020b;a). In what follows, we describe some of the approaches most connected to our work. For a more extensive survey we refer to (De Lange et al., 2021; Hadsell et al., 2020).

Two continual learning scenarios will be considered, task-incremental and class-incremental learning (Sodhani et al., 2022). In task-incremental learning, the model observes a sequence of task with known task identities. These task identities have disjoint label spaces and are provided both in training and testing. This is not the case for the class-incremental setting, where task identities must be inferred by the model to make predictions. Therefore, class-incremental learning is a considerably more challenging setting (Masana et al., 2022). Continual learning methods typically fit into one of three categories: regularization-based, memory-based (also called replay methods) or architecture-based. Regularization methods (Kirkpatrick et al., 2017; Zenke et al., 2017) augment the loss in order to prevent drastic changes in model parameters, consolidating previous knowledge. Moreover, architecture based methods (Mallya & Lazebnik, 2018) isolate or freeze a subset of the model parameters for each new observed task. In this work, we will focus on Memory-based methods, which store a small subset of the previously seen instances,(Rolnick et al., 2018; Chaudhry et al., 2018; Aljundi et al., 2019) and usually outperform their memoryless counterparts Zhou et al. (2023).

Mainly, what differentiates memory-based methods is the way in which the buffer is managed. In order to avoid forgetting, Experience Replay (Rolnick et al., 2018) modifies the training procedure by averaging the gradients of the current samples and the replayed samples. To manage the buffer, this method has two main variants: Reservoir sampling and Ring sampling. In Resevoir sampling (Vitter, 1985), a sample is stored with probability $B/N$, where $B$ is the memory budget and $N$ the number of samples observed so far. This is particularly useful when the input stream has unknown length, and attempts to match the distribution of the memory buffer with that of data distribution. The Ring strategy method prioritizes uniformity among classes, and performs class-wise FIFO sampling (Chaudhry et al., 2018).

To select the stored instances, iCARL Rebuffi et al. (2016) samples a set whose mean approximates the class mean in the feature space. During training, iCARL uses knowledge distillation and in inference, the nearest mean-of-exemplar classification strategy is performed. Some continual learning methods formulate it as a constrained optimization problem. For instance, (Aljundi et al., 2019) tries to find the subset of constraints that best approximate the feasible region of the original forgetting requirements. This is shown to be equivalent to a diversity strategy for sample selection. Another example is GEM Lopez-Paz & Ranzato (2017); Chaudhry et al. (2018), where the constrained formulation leads to projecting gradients so that model updates do not interfere with past tasks. Lastly, X-DER (Boschini et al., 2022) is a variant of (Buzzega et al., 2020b), is considered a strong baseline and uses both replay and regularization. The strategy promotes consistency with its past by matching the model's logits throughout the optimization trajectory. In (Peng et al., 2023), the general theoretical framework of ideal continual learners is analyzed, and the constrained learning formulation is put forward.

### A.2   PROOF OF THEOREM 3.2

We start by viewing the optimal value of problem $P_t$, i.e: $P_t^\star(\epsilon_k)$ as a function of the constraint tightness (or forgetting tolerance) $\epsilon_k$ associated to task $k$. Let $\epsilon = [\epsilon_1, \ldots, \epsilon_k, \ldots, \epsilon_t]$.

$$P_t^\star = \arg\min_{f \in \mathcal{F}} \; \mathbb{E}_{\mathfrak{D}_t}[\ell(f(x), y)],$$
$$\text{s.t.} \quad \mathbb{E}_{\mathfrak{D}_k}[\ell(f(x), y)] \le \epsilon_k, \quad \forall\, k \in \{1, \ldots, t\},$$

The Lagrangian $L(f, \lambda; \epsilon)$ associated to this problem can be written as

$$L(f, \lambda; \epsilon) = \mathbb{E}_{\mathfrak{D}_t}[\ell(f(x), y)] + \sum_{k=1}^{t} \lambda_k \left(\mathbb{E}_{\mathfrak{D}_k}[\ell(f(x), y)] - \epsilon_k\right) z$$

where the dependence on $\epsilon$ is explicitly shown. From Assumption 2.2, we have that problem $P_t$ is strongly dual (i.e: $P_t^\star = \max_\lambda \min_f L(f, \lambda)$). This is because it is a functional program satisfiying the convexity and strict feasiblility constraint qualification. Then, following the definition of $P_t^\star(\epsilon)$ and using strong duality, we have

$$P_t^\star(\epsilon) = \min_f L(f, \lambda^\star(\epsilon); \epsilon) \leq L(f, \lambda^\star(\epsilon); \epsilon)$$

with the inequality being true for any function $f \in \mathcal{F}$, and where the dependence of $\lambda^\star$ on $\epsilon$ is also explicitly shown. Now, consider an arbitrary $\epsilon' = [\epsilon_1, \ldots, \epsilon'_k, \ldots, \epsilon_t]$ which matches $\epsilon$ at all indices but $k$, and the respective primal function $f^\star(\cdot; \epsilon')$ which minimizes its corresponding Lagrangian. Plugging $f^\star(\cdot; \epsilon')$ into the above inequality, we have

$$P_t^\star(\epsilon) \leq L(f^\star(\cdot; \epsilon'), \lambda^\star(\epsilon); \epsilon)$$

$$= \mathbb{E}_{\mathfrak{D}_t}[\ell(f^\star(x; \epsilon'), y)] + \sum_{k=1}^{t} \lambda_k^\star(\epsilon) \left(\mathbb{E}_{\mathfrak{D}_k}[\ell(f^\star(x; \epsilon'), y)] - \epsilon_k\right)$$

Now, since $f^\star(\cdot; \epsilon')$ is *optimal* for constraint bounds given by $\epsilon'$ and complementary slackness holds, we have:

$$\mathbb{E}_{\mathfrak{D}_t}[\ell(f^\star(x; \epsilon'), y)] = P_t^\star(\epsilon').$$

Moreover, $f^\star(\cdot; \epsilon')$ is, by definition, feasible for constraint bounds given by $\epsilon'$. In particular,

$$\mathbb{E}_{\mathfrak{D}_k}[\ell(f^\star(x; \epsilon'), y)] \leq \epsilon'_k$$

This implies that,

$$\mathbb{E}_{\mathfrak{D}_k}[\ell(f^\star(x; \epsilon'), y)] - \epsilon_k$$
$$= \mathbb{E}_{\mathfrak{D}_k}[\ell(f^\star(x; \epsilon'), y)] - \epsilon_k + \epsilon'_k - \epsilon'_k$$
$$= \alpha + (\epsilon'_k - \epsilon_k) \quad \text{with } \alpha \leq 0$$

Combining the above, we get

$$P_t^\star(\epsilon) \leq P_t^\star(\epsilon') + \lambda_k^\star(\epsilon)(\epsilon' - \epsilon_k)$$

where we used that for all $i \neq k$, $\epsilon'_i = \epsilon_i$ and thus $\lambda_i^\star(\epsilon) \left(\mathbb{E}_{\mathfrak{D}_k}[\ell(f^\star(x; \epsilon'), y)] - \epsilon_i\right) \leq 0$. Equivalently,

$$P_t^\star(\epsilon') \geq P_t^\star(\epsilon) - \lambda_k^\star(\epsilon)(\epsilon'_k - \epsilon_k),$$

which matches the definition of the sub-differential, completing the proof. This result stems from a sensitivity analysis on the constraint of problem $(P_t)$ and more general versions of it are well-known in the convex optimization literature (see e.g, Bonnans & Shapiro (1998)).

### A.3 PROOF OF THEOREM 4.2

To alleviate the notation, we denote the statistical risks by $L_k(f) := \mathbb{E}_{\mathfrak{D}_k}[\ell(f(x), y)]$ and the empirical risks by $\hat{L}_k(f) := \frac{1}{n_k} \sum_{i=1}^{n_k} \ell(f(x_i), y_i)$.

As in section 4.3, $g_p(\lambda) = \min_{\theta \in \Theta} \mathcal{L}(\theta, \lambda)$ and $g_u(\lambda) = \min_{f \in \mathcal{F}} \mathcal{L}(f, \lambda)$ denote the parametrized and unparametrized dual functions. Similarly, $\hat{g}_p(\lambda) := \min_\theta \hat{\mathcal{L}}(f_\theta, \lambda)$ denotes the parametrized *empirical* dual function.

From assumption 2.4, we have that $\forall \theta \in \Theta$,

$$|L_i(f_\theta) - \hat{L}_i(f_\theta)| \leq \mathcal{O}\left(\frac{MR\sqrt{d \log(n_i) \log(|\Theta|/\delta)}}{\sqrt{n_i}}\right) := \zeta(n_i, \delta)$$

with probability at least $1 - \delta$ over a sample of size $n_i$ (see Shalev-Shwartz et al. (2009)).

From the $c-$strong concavity of $g_u(\lambda)$ in $\mathcal{B}_u$, we have that:

$$g_u(\lambda) \leq g_u(\lambda_u^\star) + \nabla g_u(\lambda_u^\star)^T(\lambda - \lambda_u^\star) - \frac{c}{2}\|\lambda - \lambda_u^\star\|^2 \quad \forall \lambda \in \mathcal{B}_u$$

Evaluating at $\hat{\lambda}_p^\star$ and using that $\nabla g_u(\lambda_u^\star) = L(f(\lambda_u^\star))$:

$$g_u(\hat{\lambda}_p^\star) \leq g_u(\lambda_u^\star) + L(f(\lambda_u^\star))^T(\hat{\lambda}_p^\star - \lambda_u^\star) - \frac{c}{2}\|\hat{\lambda}_p^\star - \lambda_u^\star\|^2$$

By complementary slackness, $L(f(\lambda_u^\star))^T \lambda_u^\star = 0$. Then, since $f(\lambda_u^\star)$ is feasible and $\hat{\lambda}_p^\star \geq 0$: $L(f(\lambda_u^\star))^T \hat{\lambda}_p^\star \leq 0$. Thus,

$$g_u(\hat{\lambda}_p^\star) \leq g_u(\lambda_u^\star) - \frac{c}{2}\|\hat{\lambda}_p^\star - \lambda_u^\star\|^2$$

Then, using Proposition 4.1, we have that:

$$\begin{aligned}
\frac{c}{2}\|\hat{\lambda}_p^\star - \lambda_u^\star\|^2 &\leq g_u(\lambda_u^\star) - g_p(\hat{\lambda}_p^\star) + M\nu(1 + \|\hat{\lambda}_p^\star\|_1) \\
&= g_u(\lambda_u^\star) \pm g_p(\lambda_p^\star) - g_p(\hat{\lambda}_p^\star) + M\nu(1 + \|\hat{\lambda}_p^\star\|_1)
\end{aligned} \tag{8}$$

Note that $g_u(\lambda_u^\star) - g_p(\lambda_p^\star) \leq 0$ since $g_u(\lambda) - g_p(\lambda) \leq 0 \quad \forall \lambda$. Therefore,

$$\frac{c}{2}\|\hat{\lambda}_p^\star - \lambda_u^\star\|^2 \leq g_p(\lambda_p^\star) - g_p(\hat{\lambda}_p^\star) + M\nu(1 + \|\hat{\lambda}_p^\star\|_1)$$

Since $\hat{\lambda}_p^\star$ maximizes its corresponding dual function, we have that $\hat{g}_p(\lambda_p^\star) \leq \hat{g}_p(\hat{\lambda}_p^\star)$. Then,

$$\begin{aligned}
\frac{c}{2}\|\hat{\lambda}_p^\star - \lambda_u^\star\|^2 &\leq g_p(\lambda_p^\star) \pm \hat{g}_p(\lambda_p^\star) - g_p(\hat{\lambda}_p^\star) + M\nu(1 + \|\hat{\lambda}_p^\star\|_1) \\
&\leq g_p(\lambda_p^\star) - \hat{g}_p(\lambda_p^\star) + \hat{g}_p(\hat{\lambda}_p^\star) - g_p(\hat{\lambda}_p^\star) + M\nu(1 + \|\hat{\lambda}_p^\star\|_1)
\end{aligned} \tag{9}$$

To conlude the proof we state the following proposition, whose proof can be found in Appendix A.4.

**Proposition A.1** *Let $\hat{\lambda}_p^\star \in \arg\max_{\lambda \succeq 0} \hat{g}_p(\lambda)$ be an empirical dual function maximizer. Under assumptions 2.2 and 2.4, we have:*

$$\begin{aligned}
|g_p(\lambda_p^\star) - \hat{g}_p(\lambda_p^\star)| &\leq 3\zeta(\tilde{n}, \delta)(1 + \|\lambda_p^\star\|_1) \qquad and \\
|g_p(\hat{\lambda}_p^\star) - \hat{g}_p(\hat{\lambda}_p^\star)| &\leq 3\zeta(\tilde{n}, \delta)(1 + \|\hat{\lambda}_p^\star\|_1)
\end{aligned} \tag{10}$$

Applying proposition A.1 to equation 9, we obtain:

$$\frac{c}{2}\|\hat{\lambda}_p^\star - \lambda_u^\star\|^2 \leq 3\zeta(\tilde{n}, \delta)(2 + \|\hat{\lambda}_p^\star\|_1 + \|\lambda_p^\star\|_1) + M\nu(1 + \|\hat{\lambda}_p^\star\|_1) \tag{11}$$

which concludes the proof.

### A.4 PROOF OF PROPOSITION A.1

Let $\hat{\lambda}_p^\star \in \arg\max_{\lambda \succeq 0} \hat{g}_p(\lambda)$ be an empirical dual function maximizer. We want to show that:

$$\begin{aligned}
|g_p(\lambda_p^\star) - \hat{g}_p(\lambda_p^\star)| &\leq 3\zeta(N, \delta)(1 + \|\lambda_p^\star\|_1) \qquad and \\
|g_p(\hat{\lambda}_p^\star) - \hat{g}_p(\hat{\lambda}_p^\star)| &\leq 3\zeta(N, \delta)(1 + \|\hat{\lambda}_p^\star\|_1)
\end{aligned} \tag{12}$$

From the uniform convergence assumption 2.4, we have that, $\forall \theta \in \Theta$:

$$|L_i(f_\theta) - \hat{L}_i(f_\theta)| \leq \zeta(n_i, \delta) \tag{13}$$

with probability at least $1 - t\delta$ over a sample of size $n_i$ (see Shalev-Shwartz et al. (2009)). Combining this with Holder's inequality, we obtain:

$$|\langle \lambda, L(f_\theta) - \hat{L}(f_\theta)\rangle| \leq \|\lambda\|_1 \|L(f_\theta) - \hat{L}(f_\theta)\|_\infty \leq \|\lambda\|_1 \zeta(\tilde{n}, \delta) \tag{14}$$

with probability at least $1 - t\delta$, where $\tilde{n} = \min_{i=1,\cdots,t} n_i$.

We will denote $\hat{f}_\theta \in \arg\min_\theta \hat{\mathcal{L}}(f, \lambda)$ an empirical Lagrangian minimizer associated to the multiplier $\lambda$ and by $f_\theta \in \arg\min_\theta \mathcal{L}(f, \lambda)$ a statistical Lagrangian minimizer associated to $\lambda$. Evaluating 14 at $(f_\theta(\hat{\lambda}_p^\star), \hat{\lambda}_p^\star)$ and $(\hat{f}_\theta(\hat{\lambda}_p^\star), \hat{\lambda}_p^\star)$ we obtain:

$$
\begin{aligned}
-\zeta(\tilde{n}, \delta)(1 + \|\hat{\lambda}_p^\star\|_1) &\leq \mathcal{L}(f_\theta(\hat{\lambda}_p^\star), \hat{\lambda}_p^\star) - \hat{\mathcal{L}}(f_\theta(\hat{\lambda}_p^\star), \hat{\lambda}_p^\star) \leq \zeta(\tilde{n}, \delta)(1 + \|\hat{\lambda}_p^\star\|_1) \\
-\zeta(\tilde{n}, \delta)(1 + \|\hat{\lambda}_p^\star\|_1) &\leq \mathcal{L}(\hat{f}_\theta(\hat{\lambda}_p^\star), \hat{\lambda}_p^\star) - \hat{\mathcal{L}}(\hat{f}_\theta(\hat{\lambda}_p^\star), \hat{\lambda}_p^\star) \leq \zeta(\tilde{n}, \delta)(1 + \|\hat{\lambda}_p^\star\|_1)
\end{aligned}
\tag{15}
$$

Re-arranging and summing the previous inequalities yields:

$$
\begin{aligned}
-2\zeta(\tilde{n}, \delta)(1 + \|\hat{\lambda}_p^\star\|_1) + \hat{\mathcal{L}}(f_\theta(\hat{\lambda}_p^\star), \hat{\lambda}_p^\star) - \mathcal{L}(\hat{f}_\theta(\hat{\lambda}_p^\star), \hat{\lambda}_p^\star) &\leq g_p(\hat{\lambda}_p^\star) - \hat{g}_p(\hat{\lambda}_p^\star) \\
2\zeta(\tilde{n}, \delta)(1 + \|\hat{\lambda}_p^\star\|_1) + \hat{\mathcal{L}}(f_\theta(\hat{\lambda}_p^\star), \hat{\lambda}_p^\star) - \mathcal{L}(\hat{f}_\theta(\hat{\lambda}_p^\star), \hat{\lambda}_p^\star) &\geq g_p(\hat{\lambda}_p^\star) - \hat{g}_p(\hat{\lambda}_p^\star)
\end{aligned}
\tag{16}
$$

Using that $f_\theta(\hat{\lambda}_p^\star)$ and $\hat{f}_\theta(\hat{\lambda}_p^\star)$ minimize the statistical and empirical Lagrangians respectively, we can write: $\hat{\mathcal{L}}(f_\theta(\hat{\lambda}_p^\star), \hat{\lambda}_p^\star) \geq \hat{\mathcal{L}}(\hat{f}_\theta(\hat{\lambda}_p^\star), \hat{\lambda}_p^\star)$ and $\mathcal{L}(\hat{f}_\theta(\hat{\lambda}_p^\star), \hat{\lambda}_p^\star) \geq \mathcal{L}(f_\theta(\hat{\lambda}_p^\star), \hat{\lambda}_p^\star)$. Which implies:

$$
\begin{aligned}
-2\zeta(\tilde{n}, \delta)(1 + \|\hat{\lambda}_p^\star\|_1) + \hat{\mathcal{L}}(\hat{f}_\theta(\hat{\lambda}_p^\star), \hat{\lambda}_p^\star) - \mathcal{L}(\hat{f}_\theta(\hat{\lambda}_p^\star), \hat{\lambda}_p^\star) &\leq g_p(\hat{\lambda}_p^\star) - \hat{g}_p(\hat{\lambda}_p^\star) \\
2\zeta(\tilde{n}, \delta)(1 + \|\hat{\lambda}_p^\star\|_1) + \hat{\mathcal{L}}(f_\theta(\hat{\lambda}_p^\star), \hat{\lambda}_p^\star) - \mathcal{L}(f_\theta(\hat{\lambda}_p^\star), \hat{\lambda}_p^\star) &\geq g_p(\hat{\lambda}_p^\star) - \hat{g}_p(\hat{\lambda}_p^\star)
\end{aligned}
\tag{17}
$$

Then, using 14 we obtain:

$$
-3\zeta(\tilde{n}, \delta)(1 + \|\hat{\lambda}_p^\star\|_1) \leq g_p(\hat{\lambda}_p^\star) - \hat{g}_p(\hat{\lambda}_p^\star) \leq 3\zeta(\tilde{n}, \delta)(1 + \|\hat{\lambda}_p^\star\|_1)
\tag{18}
$$

The same steps applied to $(f_\theta(\lambda_p^\star), \lambda_p^\star)$ and $(\hat{f}_\theta(\lambda_p^\star), \lambda_p^\star)$ yield:

$$
|g_p(\lambda_p^\star) - \hat{g}_p(\lambda_p^\star)| \leq 3\zeta(\tilde{n}, \delta)(1 + \|\lambda_p^\star\|_1)
\tag{19}
$$

which concludes the proof.

### A.5 STRONG CONCAVITY OF THE DUAL FUNCTION

**Definition A.2** *We say that a functional $\ell_i : \mathcal{F} \to \mathbb{R}$ is* Fréchet differentiable *at $\phi^0 \in \mathcal{F}$ if there exists an operator $D_\phi \ell_i(\phi^0) \in \mathfrak{B}(\mathcal{F}, \mathbb{R})$ such that:*

$$
\lim_{h \to 0} \frac{|\ell_i(\phi^0 + h) - \ell_i(\phi^0) - \langle D_\phi \ell_i(\phi^0), h \rangle|}{\|h\|_2} = 0
$$

*where $\mathfrak{B}(\mathcal{F}, \mathbb{R})$ denotes the space of bounded linear operators from $\mathcal{F}$ to $\mathbb{R}$.*

The space $\mathfrak{B}(\mathcal{F}, \mathbb{R})$, algebraic dual of $\mathcal{F}$, is equipped with the corresponding dual norm:

$$
\|B\|_2 = \sup \left\{ \frac{|\langle B, \phi \rangle|}{\|\phi\|_2} \ : \ \phi \in \mathcal{F}, \ \|\phi\|_2 \neq 0 \right\}
$$

which coincides with the $L_2-$norm through Riesz's Representation Theorem: there exists a unique $g \in \mathcal{F}$ such that $B(\phi) = \langle \phi, g \rangle$ for all $\phi$ and $\|B\|_2 = \|g\|_2$.

Regardless of the objective, the dual function $g_u$ is concave, since it is the minimum of a family of affine functions. However, in order to characterize its curvature, we need to further assume that:

1. The objective is $c_0-$strongly convex (this is is not an issue since using weight decay turns a convex objective into a strongly convex one).

2. The loss $\ell$ is $G-$smooth.

3. The constraint Jacobian $D_f L(f(\lambda_u^\star))$ is full-row rank, i.e: $\exists \ell > 0$ such that $\inf_{\|\lambda\|=1} \|\lambda^T D_f L(f(\lambda_u^\star))\|_2 \geq \ell$.

We recall that $L_k(f)$ denotes the statistical risk $\mathbb{E}_{\mathfrak{D}_k}[\ell(f(x), y)]$ associated to task $k$. Following this notation, $L_t(f)$ refers to the objective evaluated at $f$ and $L(f) = [L_1(f), \ldots, L_t(f)]$ denotes the vector of constraint values at $f$.

Let $\lambda_1, \lambda_2 \in \mathcal{B}_\lambda$ and $f_1 = f(\lambda_1), f_2 = f(\lambda_2)$. First order optimality conditions yield:

$$D_f L_t(f_1) + \lambda_1^T D_f L(f_1) = 0,$$
$$D_f L_t(f_2) + \lambda_2^T D_f L(f_2) = 0 \tag{20}$$

where $0$ denotes the null-opereator from $\mathcal{F}$ to $\mathbb{R}$ (see e.g: Kurdila & Zabarankin (2006) Theorem 5.3.1). Since $\nabla g_u(\lambda) = L(f(\lambda))$, we have that:

$$-\langle \nabla g_u(\lambda_2) - \nabla g_u(\lambda_2), \lambda_2 - \lambda_1 \rangle = -\langle L(f_2) - L(f_1), \lambda_2 - \lambda_1 \rangle \tag{21}$$

Then, by convexity of the functions $L_i : \mathcal{F} \to \mathbb{R}$, for $i = 1, \cdots, m$:

$$L_i(f_2) \geq L_i(f_1) + \langle D_f L_i(f_1), f_2 - f_1 \rangle,$$
$$L_i(f_1) \geq L_i(f_2) + \langle D_f L_i(f_2), f_1 - f_2 \rangle$$

Multiplying the above inequalities by $\lambda_1(i)$ and $\lambda_2(i)$ respectively and adding them, we obtain:

$$-\langle L(f_2) - L(f_1), \lambda_2 - \lambda_1 \rangle \geq \langle \lambda_1^T D_f L(f_1) - \lambda_2^T D_f L(f_2), f_2 - f_1 \rangle \tag{22}$$

Combining equations 21 and 22 we obtain:

$$-\langle \nabla g_u(\lambda_2) - \nabla g_u(\lambda_2), \lambda_2 - \lambda_1 \rangle \geq \langle \lambda_1^T D_f L(f_1) - \lambda_2^T D_f L(f_2), f_2 - f_1 \rangle$$
$$= \langle D_f L_t(f_2) - D_f L_t(f_1), f_2 - f_1 \rangle \tag{23}$$
$$\geq c_0 \|f_2 - f_1\|_2^2$$

where we used the first-order optimality condition and the $c_0-$strong convexity of the operator $L_t$.

We will now obtain a lower bound on $\|f_2 - f_1\|$, starting from the $G-$smoothness of $L_t$:

$$\|f_2 - f_1\|_2 \geq \frac{1}{G} \|D_f L_t(f_2) - D_f L_t(f_1)\|_2$$
$$= \frac{1}{G} \|\lambda_2^T D_f L(f_2) - \lambda_1^T D_f L(f_1)\|_2 \tag{24}$$
$$= \frac{1}{G} \|(\lambda_2 - \lambda_1)^T D_f L(f_2) - \lambda_1^T (D_f L(f_1) - D_f L(f_2))\|_2$$

Then, using that the constraint Jacobian $D_f L(f(\lambda_u^\star))$ is full-row rank

$$\|(\lambda_2 - \lambda_1)^T D_f L(f_2)\|_2 \geq \ell \|\lambda_2 - \lambda_1\|_2 \tag{25}$$

Using $G-$smoothness of $L_i$ we can derive:

$$\|\lambda_1^T (D_f L(f_1) - D_f L(f_2))\|_2 = \|\sum_{i=1}^m \lambda_1(i)(D_f L_i(f_1) - D_f L_i(f_2))\|_2$$
$$\leq \sum_{i=1}^m \lambda_1(i)\|D_f L_i(f_1) - D_f L_i(f_2)\|_2 \tag{26}$$
$$\leq \sum_{i=1}^m \lambda_1(i)G\|f_1 - f_2\|_2$$
$$= G\|\lambda_1\|_1 \|f_1 - f_2\|_2$$

Then, using the reverse triangle inequality:

$$\|(\lambda_2 - \lambda_1)^T D_f L(f_2) - \lambda_1^T (D_f L(f_1) - D_f L(f_2))\|_2$$
$$\geq \|(\lambda_2 - \lambda_1)^T D_f L(f_2)\|_2 - \|\lambda_1^T (D_f L(f_1) - D_f L(f_2))\|_2 \tag{27}$$
$$\geq \ell \|\lambda_2 - \lambda_1\|_2 - G\|\lambda_1\|_1 \|f_2 - f_1\|_2$$

Combining this with equation 24 we obtain:

$$\|f_2 - f_1\|_2 \geq \frac{1}{G} \left( \ell \|\lambda_2 - \lambda_1\|_2 - G\|\lambda_1\|_1 \|f_2 - f_1\|_2 \right)$$
$$\longrightarrow \|f_2 - f_1\|_2 \geq \frac{\ell}{G(1 + \|\lambda_1\|_1)} \|\lambda_2 - \lambda_1\|_2 \tag{28}$$

This means that we can write equation 23 as:

$$-\langle \nabla g_u(\lambda_2) - \nabla g_u(\lambda_1), \lambda_2 - \lambda_1 \rangle \geq \frac{c_0 \, \ell^2}{G^2(1 + \|\lambda_1\|_1)^2} \|\lambda_2 - \lambda_1\|_2^2$$

Letting $\lambda_2 = \lambda_u^\star$, we obtain that a lower bound of the strong concavity constant of $g_u$ in $\mathcal{B}_\lambda$ is $c = \frac{c_0 \, \ell^2}{G^2(1 + \max\{\|\lambda\| \,:\, \lambda \in \mathcal{B}_\lambda\})^2}$.

A similar proof in the finite dimensional case can be found in (Guigues, 2020).

### A.6   PROOF OF PROPOSITION 3.1

Consider the average predictor $\bar{f}(x) := \frac{1}{T} \sum_{i=1}^T f_i(x)$. Let $m_k$ be the uncosntrained minimum associated to a given task $k$. Then, we can write the expected loss of $\bar{f}$ as:

$$\mathbb{E}_{\mathfrak{D}_k}[\ell(\bar{f}(x), y)] = \mathbb{E}_{\mathfrak{D}_k}[\ell(\bar{f}(x), y)] \pm m_k \tag{29}$$
$$= m_k + \mathbb{E}_{\mathfrak{D}_k}[\ell(\bar{f}(x), y)] - \mathbb{E}_{\mathfrak{D}_k}[\ell(f_k^\star(x), y)] \tag{30}$$
$$\leq m_k + \mathbb{E}_{\mathfrak{D}_k}[M d(\bar{f}(x), f_k^\star(x))] \tag{31}$$
$$\leq m_k + \mathbb{E}_{\mathfrak{D}_k}\left[ \frac{M}{T} \sum_{i=1}^T d(f_i(x), f_k^\star(x)) \right] \tag{32}$$

Then, using that $f_i \in \mathcal{F}_i^\star$ and $f_k^\star \in \mathcal{F}_k^\star$, and that the Haussdorf distance between these two sets is bounded by $\delta$, we can write:

$$\mathbb{E}_{\mathfrak{D}_k}[\ell(\bar{f}(x), y)] \leq m_k + \frac{T-1}{T} M \delta \tag{33}$$

where we used that for $i = k$, we can set $f_i = f_k^\star \in \mathcal{F}_k^\star$ and thus, that term does not contribute to the loss.

### A.7   EXPERIMENTAL SECTION EXTENSION

At iteration $t$, models are initialized using $f_{t-1}$, except for iteration 0, where the default PyTorch implementation is used. We adopt the baseline implementations of Mammoth[2], and use their reported hyperparameters for the baselines.

To measure the final mean accuracy in Task Incremental Learning (CIL), the logits of classes not present in a given task are masked. All models are trained with SGD, using weight decay with a constant of 0.0001 and for a duration of 20 epochs in Seq-MNIST and OrganA, 30 epochs in SpeechCommands and 50 epochs in TinyImagenet.

In SpeechCommands the audio signals are resampled to 8 kHz. The PyTorch implementation of 1D convolutions is used to process these signals. The kernel sizes in this CNN are: 80 (and a stride of 16), 3, 3, 3. The original Speech Commands dataset has 35 categories, out of which 10 are used in the reduced version.

We provide additional numerical results for alternative buffer sizes 1000 and 4000 in the Speech-Commands and MNIST dataset.

---

[2] https://github.com/aimagelab/mammoth

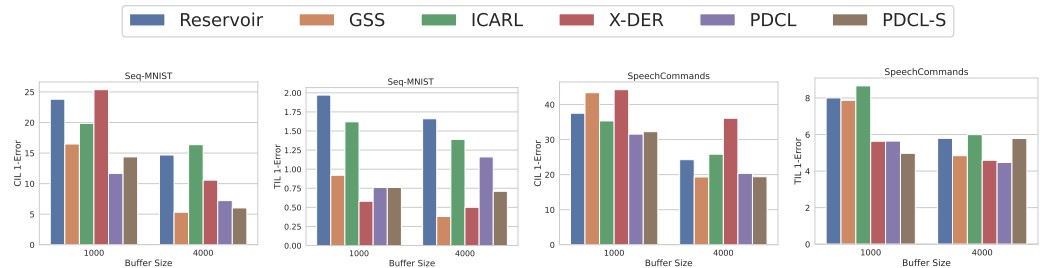

Figure 6: Mean Error accross tasks in both the task and class incremental setting in Seq-MNIST and Speechcommands for alternative buffer sizes 1000 and 4000.

## A.8 SOLVING THE BUFFER PARTITION PROBLEM

As mentioned in section 4, the buffer partition that minimizes this generalization gap can be found through the following non-linear constrained optimization problem:

$$n_1^\star, \ldots, n_t^\star = \underset{n_1, \ldots, n_t \geq n_{\min}}{\arg\min} \quad \sum_{k=1}^{t} \lambda_k \, \zeta(n_k), \tag{BP}$$

$$\text{s.t.} \quad \sum_{k=1}^{t} n_k = |\mathcal{B}|.$$

where $\zeta(n_k) = O\left(\dfrac{RM\sqrt{d \log(n_k)\log(d/\delta)}}{\sqrt{n_k}}\right)$.

Since the difficulty of a task can be assessed through its corresponding dual variable, in this problem we minimize the sum of task sample complexities, weighting each one by its corresponding dual variable and restricting the total number of samples to the memory budget. Removing the multiplicative constants that do not change the optimal buffer partition, the objective can be written as:

$$\sum_{k=1}^{t} \lambda_k \sqrt{\frac{\log(n_k)}{n_k}}$$

The aspect of the non-convex summands in the objective is shown in Figure 7.

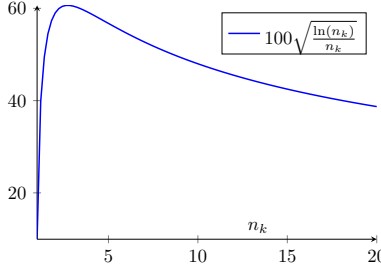

Figure 7: Aspect of $\zeta$ function.

As can be seen by analyzing the curvature of $\zeta$ (illustrated in Figure 7), if $n_{\min} = 0$ problem BP has a trivial, undesirable solution. This solution is allocating all samples to the task with highest dual variable and setting all other $n_k$ to 0. This is easily fixable, by imposing that buffer partitions should be greater than $n_{\min}$, which corresponds to number of samples $n_k$ where $\sqrt{\frac{\log(n_k)}{n_k}}$ starts decreasing. Since $\zeta$ describes a limiting behaviour, setting a lower bound on its input is reasonable. In this problem, the objective is sensical and adequately weights each sample complexity $\zeta(n_k)$ by the relative difficulty $\lambda_k^*$ of task $k$.

To solve this problem we use Sequential Quadratic Programming (SQP) Boggs & Tolle (1995), as implemented in Kraft (1988). SQP is a well known method nonlinear minimization problems with constraints. The main idea in SQP is to approximate the original problem by a Quadratic Programming Subproblem at each iteration, in a similar vein to quasi-Newton methods. An extensive description of this algorithm can be found in (Gill & Wong, 2011)

