# OpenReview forum: "Primal-Dual Continual Learning: Stability and Plasticity through Lagrange Multipliers"
_ICLR.cc/2024/Conference — Submitted to ICLR 2024_

### Official Review · Reviewer_aK2g · 2023-10-22

**Soundness:** 3 good
**Presentation:** 3 good
**Contribution:** 3 good
**Rating:** 8
**Confidence:** 4

**Summary:**

This paper formulates the no-forgetting objective of Continual-Learning (CL) as a constrained optimization problem w.r.t the population risks. Given the forgetting tolerance $\epsilon_{1:T}$, it focuses on two important aspects of the memory-based methods: 1. how to partition the memory buffer for different tasks. 2. For each task, which subsamples should be stored? The first point is addressed by deciding the sample size of each task through minimizing the generalization gap weighted by the optimal dual variables of the CL objective. The second is to select the samples with the highest associated per-sample dual variable from each task.

**Strengths:**

1. The paper is well-written and the motivation is clear.
2. Relating the generalization gap with the dual variables to obtain the optimal memory partition in CL is novel to me.
3. Experimental results validate the effectiveness of the proposed method compared to previous memory-based approaches.

**Weaknesses:**

My primary concerns lie in the following aspects:
  * The convergence of $\mathbf{\lambda}$ is highly sensitive to the setting of the forgetting tolerance $\epsilon$, the number of tasks $T$, and the hardness of the tasks, which will affect the memory partition.
  *  At every timestep, the memory partition changes. Not just the problem mentioned in the discussion exists, where the optimal partition size of a previous task grows at the current timestep. For the tasks that have a smaller size at the current timestep, it needs to reselect the samples to store, which would cause additional computation costs.
  * The growing and large number of constraints.

**Questions:**

Please see the previous section.

---

> ### Author Response · Authors · 2023-11-20
> **Response to reviewer aK2g**
>
> We sincerely thank the reviewer for their time and feedback, and for their positive evaluation of our work.
>
> - Sensitivity of $\lambda$
>
> We believe that the sensitivity of $\lambda$ with respect to the forgetting tolerance $\epsilon$ and the hardness of the tasks is indeed a *positive* aspect of our approach. As mentioned in the general response, the regularization parameter associated to each rehearsal loss is meant to be adaptive. For instance, if the forgetting tolerance $\epsilon_k$ associated to task $k$ is relaxed, the effective difficulty of this task is reduced, decreasing its associated dual variable and leading to a smaller buffer partition. This adaptivity is the insight leveraged to partition the buffer and select samples.
>
> - Re-computation of buffer partition.
>
> Indeed, the buffer partition is computed at every iteration, which creates an additional cost with respect to fixed partitions. However, the computational overhead of running the Sequential Quadratic Programming algorithm is small (see Appendix A.8). More importantly, as mentioned in the previous response, the flexibility of recomputing the buffer partition after every iteration allows us to dynamically assess the relative difficulty of the observed tasks, which is a key positive aspect of the proposed approach.
>
> As mentioned in Section 7, there are two non-efficient ways of addressing the problem of dual variable underestimation. Namely, one can fill the empty portion of the buffer with either augmented samples from the previously selected ones or samples from the current task, whose dataset is entirely available.
>
> - Number of constraints
>
> The number of constraints in our formulation of continual learning can be equal to the number of tasks or the number of samples. In the sample-level setting, our experiments show that one can effectively train constrained continual learners with up to 100000 constraints (Tiny-ImageNet). In view of the challenges and novel techniques deployed in this paper, we leave the experimental validation on larger datasets (e.g., ImageNet) as future work.

---

> > ### Comment · Reviewer_aK2g · 2023-11-21
> >
> > Thanks for your clarification.
> >
> > My concerns are partly resolved. Overall, even though it's not as perfect as I expected, I believe it's an interesting work that should be seen by the community.
> >
> > I raised my score accordingly.

---

### Official Review · Reviewer_CG5c · 2023-10-30

**Soundness:** 3 good
**Presentation:** 3 good
**Contribution:** 3 good
**Rating:** 6
**Confidence:** 3

**Summary:**

This paper views continual learning as a constrained learning problem: to learn the new task without forgetting the old tasks (too much). Some previous work took this perspective as well, but in those cases this way of formulating the continual learning problem only motivated the proposed approach. In this paper, the authors directly address continual learning as a constrained learning problem by making use of recent advances in Lagrangian duality as tool address constrained optimization. In particular, the paper demonstrates that by adopting such a primal-dual method, a principled approach emerges for deciding how to fill the memory buffer.

**Strengths:**

As far as I am aware, this is the first work that directly addresses continual learning as a constrained learning problem. The paper proposes a principled framework for this by means of optimizing the Langrangian empirical dual, and it provides clear theoretical justification for its propositions.

A neat theoretical demonstration of the paper is showing that the Lagrangian dual variables can be interpreted as signaling the difficulty of their corresponding task.

The paper then demonstrates that the Lagrangian dual variables can be used to select which samples to store in the memory buffer, and that empirical benefits can be obtained by doing so.

**Weaknesses:**

Although I think this paper already makes some important and neat contributions, to realize its full potential, I think it is important to improve and clarify the empirical comparisons.

**Indirectness of empirical comparisons**

In my opinion, from a practical perspective, this paper proposes three “novel aspects” compared to the standard experience replay approach that is commonly used in continual learning:

{1} the weighing of the replayed losses relative to the loss on the current task is determined by the Lagrangian dual variables (rather than, as is currently done in continual learning, either by a hyperparameter or as a function of how many tasks have been seen so far)

{2} the selection of samples to be stored in the buffer at the task level (buffer partition)

{3} the selection of samples to be stored in the buffer at the sample level

However, it seems only the impact of the last two aspects are evaluated empirically. Why do the authors not include a direct comparison to assess the effect of {1}? (That is, a comparison between "standard ER" and the approach proposed by this paper except without buffer partition at task level or individual sample selection.) I think doing so could substantially strengthen this paper. Moreover, it is not clear to me whether the comparisons to assess the effect of {2} are direct. For example, in Figure 1 (but a similar question applies to Figure 4), when “PDCL” is compared with “Reservoir”, it is not clear how the replayed losses are weighed in the case of “Reservoir”. Are they weighed in the same way as in “PDCL”? Or are they weighed in another way? This should be clearly described. If it is the second option, then I do not think that Figure 1 provides a comparison that “isolates the effect of buffer partition”.

**Distinction task- versus class-incremental learning**

The way the paper describes the difference between task- and class-incremental learning suggests that the authors *train* their models in these two scenarios in the same way, and that there is only a difference between these scenarios in the way the models are *evaluated* at test time. Is this indeed the way the authors implemented their experiments? Because to me it seems there should also be a difference in how models are trained in task-incremental versus class-incremental learning. For example, when training on samples from the second task, with task-incremental learning the models only need to be trained on distinguishing between classes from the current task, while with class-incremental learning the models should also learn that those current samples do not belong to classes from the first task. To clarify this, the authors should provide more details regarding how they implemented the difference between task- and class-incremental learning. When discussing the distinction between task- and class-incremental learning, I think it is also important to cite the original paper (van de Ven et al., 2022; https://www.nature.com/articles/s42256-022-00568-3).

**Minor issues:**
- top of p9: a reference is made to Figure 9, but I think Figure 4 might be meant?
- in the reference list, the paper Buzzega et al. (2020) is included twice
- for a number of papers in the reference list, no venue is included (e.g., Gentile et al., 2022; but there are several others as well)
- there are several formatting issues with in-text citations in the Appendix
- on p19, Task Incremental Learning is abbreviated as CIL

**Questions:**

Although I think this paper already makes some important and neat contributions, to realize its full potential, I think the authors should [1] include empirical comparisons that more directly assess the impact of the three different novel aspects that the authors propose, and [2] provide more details regarding the difference between the task- and class-incremental learning experiments.

Please see under “Weaknesses” for details on both.

While I think it is already a paper that could be accepted, if these two issues can be satisfactorily addressed, I think it could be a strong or very strong paper.

I would be happy to actively engage in the discussion period.

---

> ### Author Response · Authors · 2023-11-20
> **Response to reviewer CG5c**
>
> We thank the reviewer for their feedback and for the positive evaluation of our work.
>
> - On the indirectness of empirical comparisons.
>
> Indeed, the effect of adaptive weights for replay losses (as opposed to fixed hyperparameters) is evaluated only indirectly, along with the buffer partition strategy. We agree with the reviewer in that adding the explicit comparison against a primal-dual method that uses the same buffer partition as ER is pertinent (we refer to it as PD w. Reservoir). We provide the preliminary results of this comparison here (CIL setting) and will add the full experiments to the revised version. The main conclusion appears to be that adaptive weights alone provide an improvement with respect to fixed ones. Indeed, in the case of SpeechCommands with a buffer size of 2000, virtually all the gains are explained by the adaptivity of the task-loss weights. However, in many cases, this does not explain the performance improvement entirely.
>
>
> | Test Accuracy  | SeqMNIST |       | SpeechCommands  |       |
> |-------------|-------|-------|-------|-------|
> | Buffer Size | ER   | PD w. Reservoir   | ER   | PD w. Reservoir   |
> | 400         | 70.28 | 82.1 | 45.75 | 49.43 |
> | 2000        | 87.02  | 88.97 |  68.54 | 72.28 |
>
>
> Regarding the baseline Experience Replay, we use the standard implementation from (Chaudhry et al, 2019). At each iteration, a batch of samples from the buffer and a batch of samples from the current task are merged. This larger batch is used to perform a forward pass and to compute the gradient of the loss with respect to $\theta$. Thus, the relative weight associated to each task loss is proportional to the number of samples of that task available in the buffer. For instance, when using a uniform partition, the relative weights are also uniform.
>
> Chaudhry, A., Rohrbach, M., Elhoseiny, M., Ajanthan, T., Dokania, P.K., Torr, P.H. and Ranzato, M.A., 2019. On tiny episodic memories in continual learning. arXiv preprint arXiv:1902.10486.
>
> - Distinction task versus class incremental learning.
>
> We follow the formulation of class and task incremental learning of (Boschini et al. 2022), as implemented by the continual learning library Mammoth (https://github.com/aimagelab/mammoth). In this implementation, only the evaluation procedure is affected. The reviewer is correct in that this implementation provides an approximation to the TIL setting since it does not affect the training procedure. When evaluating the model in the TIL setting, only the relevant classes are taken into account. In CIL, however, the predictor can select any of the previously observed categories. We will clarify this distinction in the revised version. We will add the citation to the mentioned reference (van de Ven et al., 2022) and clarify this in the revised version.
>
> Boschini et al, 2022. Class-Incremental Continual Learning into the eXtended DER-verse
>
> - Other comments
>
> Thank you very much for pointing these out. We will correct the formatting issues and appendix references in the revised version.

---

> ### Comment · Reviewer_CG5c · 2023-11-22
>
> Thank you for responding to my review. I appreciate the additional experiment in which a direct comparison is performed between "standard ER" and "adaptive weights for replay". I note that the outcome of this experiment seems to challenge, at least to a certain degree, the conclusions from the original submission regarding the benefit of buffer partitioning.
>
> Although I think that a difference in performance between "standard ER" and "adaptive weights for replay" would also be an interesting finding, I think it is unfortunate that based on the currently communicated results it is unclear which conclusions can be drawn from the experiments.
>
> By proposing a principled framework for directly addressing continual learning as a constrained optimisation problem, I still think this paper makes an interesting and insightful contribution, and therefore has enough merit to be accepted. However, the current incompleteness of the results, and the resulting ambiguity with regards to what conclusions the authors will be able to draw, prevent me from raising my score further.
>
> Further, reading the other reviews and the author rebuttals to them, I was surprised that the authors seem to find that A-GEM performs substantially better than "standard ER". (Comparing the results reported in the rebuttal to reviewer "Heyc" with the results reported in the rebuttal to my review.) I have not come across experiments before in which A-GEM clearly outperforms "standard ER". Based on my reading of the literature, it seems to be usually the other way around (e.g., De Lange et al., 2022; https://ieeexplore.ieee.org/abstract/document/9349197 or Van de Ven et al., 2022; https://www.nature.com/articles/s42256-022-00568-3). I think it would be good if the authors double-check their results, and if they are indeed correct, it would be useful to discuss why their results might differ from the ones in the above articles.

---

> ### Author Response · Authors · 2023-11-22
>
> We thank the reviewer for their comments and acknowledgment of the contribution of the principled constrained learning framework. We are in agreement about the benefits of analyzing a Primal Dual method with no buffer partition to further assess the impact of adaptive weights. Regarding the extra baseline A-GEM. Indeed, it is well-known that the original version of A-GEM is outperformed (in many cases by a large margin) by ER, both with reservoir and ring buffer, as reported in: [On Tiny Episodic Memories in Continual Learning. Chaudhry et al. 2019]. Thus, it is not uncommon to use A-GEM gradient updates but to augment its loss with a replay term. This can be thought of as a mixture of strategies (AGEM with ER), present in some Continual Learning code bases, and recently formalized in [Two Complementary Perspectives to Continual Learning: Ask Not Only What to Optimize, But Also How. Hess, Tuytelaars, van de Ven]. We apologize for the confusion and will revise our response to clarify that this is the version of A-GEM used.

---

> > ### Comment · Reviewer_CG5c · 2023-11-22
> >
> > Thank you for the clarification regarding A-GEM. That makes sense to me.

---

### Official Review · Reviewer_ipX4 · 2023-10-30

**Soundness:** 2 fair
**Presentation:** 2 fair
**Contribution:** 2 fair
**Rating:** 3
**Confidence:** 3

**Summary:**

This work directly leverages the constrained optimization framework to solve a continual learning problem.
Based on the renowned sensitivity analysis with Lagrangian dual variables, this work tackles the continual learning problem in two different aspects, at the level of tasks and data.
* At the task level, the Primal-Dual Continual Learning (PDCL) algorithm allocates more datapoints to task that is sensitive to constraint perturbation (i.e. large per-task dual variable)
* At the data level, their indirect sample selection algorithm prefers to choose datapoints that are sensitive to constraint perturbation (i.e. large per-datum dual variables)

**Strengths:**

* The authors carefully motivate the readers to understand Lagrangian sensitivity analysis and its application in the context of continual learning.
* Their experiments show that the idea of Lagrangian sensitivity analysis can be effectively applied to both buffer allocation and data selection for memory-based continual learning.

**Weaknesses:**

* **Regarding theoretical contributions**
  - In abstract, it is claimed that there are sub-optimality bounds. At first glance, I was expecting learnability guarantee (e.g., PAC) for the actual continual learning problem. However, it turns out that the sub-optimality bound was for estimation of dual variables. Since the estimation of dual variables is the main spirit of the proposed algorithm, I don’t want to say this is not an enough contribution. Rather, I would say the expression ‘sub-optimality bound’ is quite misleading in some sense.
  - The paper defers the discussion on the *strong* concavity constant $c$ to Appendix A.5. However, I think this hides several important dependencies. For example,
    1. It intrinsically assumes the usage of (might be a large amount of) weight decay to induce strong concavity of the objective function;
    2. The loss function should be $G$-smooth, and the sub-optimality bound in Theorem 4.2 turns out to depend quadratically on $G$.
    3. The constraint Jacobian (question: what is it exactly?) must be full rank, and the sub-optimality bound depends quadratically on the inverse of the minimum singular value of this matrix, which can be arbitrarily large.

    For these, I think the paper should be more clear and honest on several hidden dependencies.
  - The last paragraph of Section 4 claims that the weakness of the sub-optimality bound “can be fixed by replacing the minimum with the average sample complexity”, but I cannot find any detailed discussion on this, throughout the paper.
  - Although the proof would be similar to that of Theorem 3.2, I think the full proof of Proposition 5.1 should be added, or at least a set of necessary modifications in the proof to prove the proposition must be added.
* **Regarding Theorem 3.2 and the notation “$\partial P^{\star}_t (\epsilon_k)$”**
  - Is $P^{\star}_t (\cdot)$ a convex function? I think this should be clarified in order to use the notion of sub-differential.
  - Also, I think the notation is quite confusing. I would like to suggest the notation like “$\partial_{\epsilon_k} P^{\star}_t (\epsilon)$” where $\epsilon = (\epsilon_1, …, \epsilon_t)$.
  - In a higher level of discussion, does the paper ever require such a **local** sensitivity result to give a motivation?
* **There are several but minor typos and misleading usages of symbols:**
  - Equation $(P_t)$: I think this should be $\min_{f\in\mathcal{F}}$, not $\arg\min_{f\in\mathcal{F}}$. This also applies to the equation at the beginning of Appendix A.2.
  - In Assumption 2.1, $\delta$ is used for task similarity. Throughout Section 4, however, $\delta$ is used as a probability parameter.
  - Assumption 2.4: “There exists $R, M >0$ such that …”
  - Page 3, below Equation $(1)$: “… two-player gamer …” $\rightarrow$ “… two-player game …”
  - Equation $(3)$: Why do we need an inner product between two scalars $-\lambda_k^\star$ and $\gamma$? I don’t think this is necessary.
  - Proposition 4.1: The order 2.3 and 2.2 must be flipped.
  - Theorem 4.2: “$\\|\lambda’\\|_1 = \max\\{\\|\lambda_p^\star\\|, \\|\hat{\lambda}_p^\star\\|\\}$” $\rightarrow$ Are all the norms $\ell_1$-norms?
  - Page 7, below Equation $(6)$: “$\mathfrak{B}_t(x,y) \ne D_t(x,y)$” $\rightarrow$ “$\mathfrak{B}_t(x,y) \ne \mathfrak{D}_t(x,y)$”
  - Section 6: there are some inconsistencies of using the word “Tiny-ImageNet”, which should be fixed throughout the section.
  - Appendix A.2, page 15, the equation starts with $L(f,\lambda;\epsilon)$: What is $z$ at the end of the equation? I think it should be removed.
  - Appendix A.5: the letter $\ell$ is both loss function and the minimum singular value of the constraint Jacobian matrix.
* **Minor comments**
  - Around Assumption 2.3, it would be great if the authors put some citations on universal approximation results for neural networks, which explains (with examples) the richness of (modern) machine learning model parametrization.
  - Below Equation $(1)$: “… the forgetting tolerances $\\{\epsilon_k\\}$ need to …” $\rightarrow$ “… the forgetting tolerances $\\{\epsilon_k\\}$ *suffice* to …”
  - Page 4: This sentence is quite weird: “… it is sensible to partition the buffer across different tasks as an increasing function of $\mathbf{\lambda}^\star$...”, because it says we can say that a function is increasing in terms of a vector variable.

Overall, I believe the writing could be much more improved than the current draft.

**Questions:**

Please see **Weaknesses**.

---

> ### Author Response · Authors · 2023-11-20
> **Response to reviewer ipX4**
>
> We thank the reviewer for their time and detailed feedback. In what follows, we provide answers and clarifications to the weaknesses and questions raised.
>
> - Regarding Theorem 3.2/5.1 and the sub-differential $\partial P^*_t(\epsilon_k)$.
>
> In this setting, the perturbation function $\partial P^{\star}\_t(\epsilon_k)$ is a convex function of the constraint upper bounds $\epsilon$. This is a well-known result from convex optimization (see e.g., (Bonnans & Shapiro, 1998) or (Rockafellar, 1997, Theorem 29.1)), and we agree with the reviewer in that it is pertinent to mention it before Theorem 3.2. We will also update the notation of the subdifferential to $P^*_{\epsilon_k}(\epsilon)$ as suggested to avoid confusion.
>
> One could motivate the buffer partition approach without this sensitivity result. Note that dual variables accumulate the slacks (task-level constraint violations) over the continual learning procedure. Thus, it is not surprising that they carry information about task difficulty and that they can be used to manage the buffer. However, Theorem 3.2 provides a complete description of the induced task difficulty metric. Theorem 3.2 tells us that dual variables capture the minimum rate of degradation of the performance on the current task, when tightening the forgetting requirement in a previous task. This translates into a principled measure of how hard maintaining the performance in a past task is, relative to other observed tasks. Thus, we believe that Theorem 3.2 is a key component of this paper. We also provide an experimental interpretation of Proposition 5.1 (see Figure 3.b).
>
> - On the estimation of dual variables.
>
> Since we only have access to samples (not probability distributions) and we optimize over model parameters (not functions), in practice, we solve the empirical problem $(\hat{D}_t)$ instead of $P_t$. This corresponds to analyzing estimation and approximation errors of constrained learning, as done by e.g., (Vapnik 1999) or (Shai-Shalev Schwarz 2014) for supervised learning. For constrained learning, these errors were characterized, to a certain extent, in (Chamon et al. 2023). Theorem 4.2 bridges the gap between the optimal dual variables of $(P_t)$ and $(\hat{D}_t)$.
>
> For the distance between these dual variables to be bounded, the dual function $g_u$ can not be flat. Observe that $g_u$ is the minimum of a family of affine functions; thus, it is concave, irrespective of whether the primal problem is convex. For it to be strongly-concave, we require three extra assumptions. Namely, strong convexity and smoothness of the loss and full-rankness of the constraint Jacobian.
>
> The first two (strong convexity and smoothness) are widely used to prove convergence results for gradient descent algorithms even in the convex setting and are satisfied by typical losses used in ML when, e.g., the hypothesis class $\mathcal{F}$ is compact. Note that we require convexity of the objective with respect to the functionals, but not model parameters, which holds for both mean squared error and cross-entropy loss.
>
> Although harder to guarantee, Assumptions 3.7 is a constraint qualification called LICQ (Linear Independence Constraint Qualification), which is ubiquitous in constrained optimization (Bertsekas 1995). The independence of the columns of $D_f \ell (f^*)$ reflects that, at the optimum, the constraints are not redundant and that they contribute in different ways to reach the optimal point. Although this is probably the stricter assumption, we believe that it can be lifted, being replaced by a stronger version of strict feasibility, but this is not immediate. With these assumptions, we can guarantee that $g_u$ is not linear, and that the optimal dual variables of $(P_t)$ and $(\hat{D}_t)$ are not too far away.
>
> Although improvements to this bound could be obtained, we leave them for future work. We believe that the current version of the bound theoretically grounds the use of dual variables as sensitivity indicators.
>
>
>  - Other comments
>
> We agree with the reviewer regarding the lack of comments on how the proof of Theorem 3.2 relates to the proof of Proposition 5.1 (i.e., sample-level sensitivity). These modifications are very minor since the only difference between Theorem 3.2 (task-level sensitivity) and Proposition 5.1 (sample-level sensitivity) is the dimensionality of the input to $P^*_t(\epsilon)$. We will add these comments to the revised version. We will also fix the typos mentioned in the revised version and add citations on universal approximation results for neural networks (e.g., Hornik 1989).
>
> Hornik, K., Stinchcombe, M. and White, H., 1989. Multilayer feedforward networks are universal approximators. Neural networks, 2(5), pp.359-366.
>
> Chamon, Luiz FO, et al. "Constrained learning with non-convex losses." IEEE Transactions on Information Theory 69.3 (2022): 1739-1760.

---

> > ### Author Response · Authors · 2023-11-20
> > **Additional references**
> >
> > J. Frédéric Bonnans and Alexander Shapiro, Optimization Problems with Perturbations: A Guided Tour. SIAM Review, Society for Industrial and Applied Mathematics.
> >
> > Rockafellar, R Tyrrell, Convex analysis, 1997. Princeton university press.
> >
> > Dimitri P. Bertsekas. Nonlinear Programming: 3rd Edition, Athena Scientific.

---

### Official Review · Reviewer_Heyc · 2023-11-03

**Soundness:** 2 fair
**Presentation:** 3 good
**Contribution:** 2 fair
**Rating:** 3
**Confidence:** 5

**Summary:**

This paper proposes a theoretical analysis of memory-based continual learning based on the recent advances in constrained optimization.

In terms of constrained optimization, preventing forgetting previously learned tasks becomes the constraint of the optimization problem, and the emprical risk with finite samples should be bounded by the forgetting tolerance as the constraints.

Motivated by the theoretical result of the constrained learning through Lagrangian duality (Chamon et. al. 2020), the authors provide a theoretical plausible Lagrange multiplier $\lambda_k$ and the buffer size $n_k$  for each task $k.$

In the experiment, the paper provides some toy benchmark results, such as seq-MNIST with several memory-based baselines.

**Strengths:**

In the research of continual learning, there is few optimization-based analysis to mitigate catastrophic forgetting.

This paper provides a new theoy-based algorithm from scratch, which helps to understand which Lagrange multiplier  and buffer size are used totrain the neural networks for continual learning.

**Weaknesses:**

Despite the theoretical result, the proposed algorithm does not fit the online continual learning scenario because the process "fill buffer" is done after visiting samples in line 11 of Algorithm 1.

This implies that the buffer should keep all encountered data points during $n_{iter}$ iterations, and then the buffer drops some samples to satisfy the buffer size condition, which has already been violated in lines 5-10 in Algorithm 1.

It seems that this contradiction occurs because Algorithm 1 needs to access the information of $\lambda_k$ at the end of each task to compute the optimal buffer size. However, we should have at least the upper bound of the buffer size for the current task $k$ to save encountering samples in the online stream.

In addition, the loss landscape on the parameter $\theta$ is non-convex, as the authors stated in Section 3. The local-optimal setting for a given local minimal point and the Lagrange multiplier do not guarantee remarkable performance in the empirical result. The existing heuristic methods based on constrained optimization, such as A-GEM, have already shown remarkable performance in more complex benchmarks, such as split-CIFAR100 and split-MiniImagenet.

Considering the recent advances in continual learning, I think that a new constrained optimization-based CL algorithm should be either theoretically solid or empirically outstanding.

**Questions:**

1. The reported metric is not standard in continual learning. Can the authors report the experiemntal result in terms of the average test accuracy and FWT?
2. I think the constrained optimizaiton based CL baselines, such as GEM and A-GEM should be included in the experiemt section to analyze the novelty of the proposed method. Is there any reason why the authors does not contain these algorithms?

---

> ### Author Response · Authors · 2023-11-20
> **Response to reviewer Heyc**
>
> We greatly appreciate the reviewer's feedback. We strongly encourage them to read our general response to all reviewers as it will clarify the main motivations and contributions tackled by our work. We hope that our answers below will clarify specific points that might have been missed.
>
> - On the availability of samples during training.
>
> The notation used could have led to some confusion regarding the availability of samples during the execution of the Primal-Dual iterations and buffer partition. As explained in section 2, at iteration $t$, the following are available:
>
> 1. Samples associated to task $t$.
>
> 2. Samples from tasks $1, \dots, t-1$ that where stored in the buffer: $\mathcal{B}\_t = \cup_{k=1}^{t-1} \mathcal{B}\_t^k$  where  $\mathcal{B}_t^k$ denotes the subset of the buffer allocated to task $k$ at iteration $t$.
>
> This is the standard memory-based continual learning setting [1], where one can store a subset of the samples from previous tasks, revisiting them at future iterations. We will clarify this in the algorithm.
>
> [1] GM van de Ven, Three types of incremental learning, 2022.
>
> - On the computation of $\mathbf{\lambda}$ at each iteration.
>
> Indeed, at each iteration $t$, we undertake the dual problem $\hat{D}_t$, from which we obtain a vector of dual variables $\mathbf{\lambda}$. This means that we assess the relative difficulty of the observed tasks and compute a new buffer partition at each iteration. However, we do not require access to the entire dataset of a previous task to compute this partition. Instead, it is estimated using the set of samples $\mathcal{B}_t$ stored in the buffer. As mentioned in section 7, this can give rise to the issue of dual variable underestimation.
>
> - On the sub-optimality of $\theta^*$.
>
> Indeed, the loss landscape on the parameter $\theta$ is non-convex, creating a duality gap. In other words, the value of the dual problem $\hat{D}^{\star}_t$ need not be a good approximation of the value of the primal problem $P^\star_t$. However, as mentioned in section 3, this was tackled in (Chamon et al, 2022, Theorem 1):
>
> $$ | P^\star_t - \hat{D}^{\star}_t | \leq (M\nu+\zeta)(1+ \Delta ) \text{,}$$ where $\Delta= \text{max} ( \Vert \tilde{\lambda}^* \Vert, \Vert \hat{\lambda^*} \Vert )$ and $\zeta = \text{max} \zeta_i(N_i, \delta)$ (see section 4.2 for the definition of $\zeta_i$ ).
>
> Constrained Learning with Non-Convex Losses, 2022. Luiz F. O. Chamon, Santiago Paternain, Miguel Calvo-Fullana, and Alejandro Ribeiro.
>
> - On the reported metric.
>
> We report the Top-1 Error, which is simple 100-Average Test Accuracy. We report this both in the Task Incremental Setup and Class Incremental Setup, both of which are standard in the continual learning literature. For completeness, we will add accuracy to the revised version in the Additional Experiments Appendix.
>
>
> - On the baselines.
>
> We agree with the reviewer that A-GEM is a relevant baseline. We attach the results of A-GEM on SpeechCommands here and will include them on the rest of the datasets in the camera-ready version. A-GEM appears to perform worse than PDCL on the Task Incremental setting (~2%), and slightly better on the Class Incremental setting.
>
> |             | A-GEM |       | PDCL  |       |
> |-------------|-------|-------|-------|-------|
> | Buffer Size | TIL   | CIL   | TIL   | CIL   |
> | 400         | 90.07 | 55.75 | 92.94 | 54.73 |
> | 2000        | 92.5  | 72.54 | 94.79 | 72.36 |

---

> ### Comment · Reviewer_Heyc · 2023-12-01
>
> Thank you for the thoughtful response.
> I have further questions to clarify as follows.
>
>
> ***On the availability of samples during training***
>
> I have already understand that the main procedure of memory-based continual learning, so my main concern was that it is not clear how the data points of the current task $t$ stores in the buffer $B_t^t$ in Algorithm1.
> More specifially, the inner loop iterates $n_{iter}$ iterations on task $t$, which might allow the multiple epochs for task $t$ by the definition. After the inner loop, the samples that will be stored are chosen by the partition rule in line 11.
> It seems that some weird points can be arised as follows.
>
> - After fininsing to learn task $t$, the algorithm tries to store data points for task $t$, which have already been streamed.
> - First, multi-epoch learning is not a standard protocol for continual learning (even in the stage of a certain task).
> - Second, we cannot access the streamed data after line 9, because the allowed memory size should be smaller than the size of dataset $D_t$.
>
> By the above reason, building buffer at $t$ in line 11 does not make sense.
> I think the author should collect some data points in the inner loop with the memory limitation condition, then compute the buffer partition with a modified rule.
>
> After reading the response, the computation $\lambda_t$ seems fair, but the authors should correct the statement of alogirhtm.
>
>
> As far as I understand, there exists a ***critical mismatch*** in the experiment setting.
> GSS and Reservoir sampling are CL algorithms for online continual learning, but the experiments are conducted on the multi-epoch setting(Section A.7). This is not a fair comparison with baselines.
>
> In addition, I still think that the author should report FWT and average ACC simultaneously in Figure 4.

---

### Author Response · Authors · 2023-11-20
**General Response**

We genuinely thank all the reviewers for taking the time to read the manuscript and provide invaluable feedback on our work. In this comment, we would like to clarify an important point regarding the main focus of our work.

The main point of this work is to show that a constrained learning formulation of continual learning can be solved through a dual sub-gradient ascent algorithm, and that it is beneficial to leverage the dual variables obtained as a by-product to manage the memory buffer efficiently.

As noted by reviewer CG5c, this perspective entails significant differences that reflect ["important and neat contributions."](https://openreview.net/forum?id=GicZtgSlJW#:~:text=Although%20I%20think%20this%20paper%20already%20makes%20some-,important%20and%20neat%20contributions,-%2C%20to%20realize%20its%20full%20potential%2C%20I%20think%20the)

1. The regularization parameter associated to each rehearsal loss is adaptive.  As shown in Algorithm 1 and explained in Section 3,  the weights associated to losses in previous tasks (i.e., dual variables), are updated during the training procedure according to the degree of constraint satisfaction or violation. Despite its simplicity, this is a significant difference from standard knowledge distillation and rehearsal approaches.

2. The use of dual variables is ["carefully motivated"](https://openreview.net/forum?id=GicZtgSlJW#:~:text=The%20authors-,carefully%20motivate,-the%20readers%20to) (as per reviewer ipX4) since they capture the stability plasticity-tradeoff in continual learning. This is a novel measure of task difficulty, and, more importantly, a useful one. We show that it provides benefits in buffer partition and some sample selection scenarios, and that, in the latter, it corresponds to sampling points close to the decision boundary (see Figure 3.b).

---

### Meta-Review · Area_Chair_54HF · 2023-12-07

**Metareview:**

The paper puts forward a constrained-optimization view on continual learning. Specifically, the constraints ensure that the loss on previous tasks is smaller than some epsilon, preventing forgetting of past knowledge.  The constrained optimization problem is solved using an inexact dual-ascent method. A connection between the Lagrange-multipliers and plasticity/stability is discussed.  The primal-dual continual learning algorithm is evaluated on some small scale experiments.

The strength of the paper is the innovative and neat idea to use duality to tackle continual learning.  The main weakness, also pointed out by reviewers, is the small-scale numerical evaluation and experimental methodology.

Moreover, there are some doubts regarding the complete soundness of the theoretical framework. These have been addressed in the rebuttal, but another round of reviews will be required to make sure these are fully resolved. Additionally to that, it remains unclear on how much of the mathematical rigour is actually needed for the practical implementation. The heavy use of variational analysis tools (e.g., Rockafellar's book) may perhaps obfuscate the main message of the paper.  For a revised version, I encourage the authors to focus on improving the numerical and experimental methodology and keeping the theory in the main text at the bare minimum required to implement and intuitively understand the method.  Details analytical proofs may deter from the main story, and I recommend to defer them to the appendix.

I encourage the authors to resubmit taking the above feedback, and the reviewers' feedback into account.

For this ICLR, I recommend to reject the paper.

**Justification For Why Not Higher Score:**

The empirical evaluation is lacking (no evaluation on standard continual learning benchmarks, no comparison to recent methods) and the experiments are rather small-scale.  The theory looks neat, and building on rather advanced optimization theories, but the need for so much mathematical rigour for the final algorithm is unclear.  For a higher score, I would have liked to see larger experiments and an improved presentation focusing solely on the theoretical concepts which are really needed, giving more space for explaining the principles at work and situating the paper in the larger literature.

**Justification For Why Not Lower Score:**

N/A

---

### Decision · Program_Chairs · 2024-01-16

Reject